# Quantification of sporozoite expelling by *Anopheles* mosquitoes infected with laboratory and naturally circulating *P. falciparum* gametocytes

Chiara Andolina[1†], Wouter Graumans[1†], Moussa Guelbeogo[2], Geert-Jan van Gemert[1], Jordache Ramijth[1], Soré Harouna[2], Zongo Soumanaba[2], Rianne Stoter[1], Marga Vegte-Bolmer[1], Martina Pangos[3], Photini Sinnis[4], Katharine Collins[1], Sarah G Staedke[5], Alfred B Tiono[2], Chris Drakeley[6], Kjerstin Lanke[1], Teun Bousema[1,6]*

[1]Department of Medical Microbiology, Radboud University Nijmegen Medical Centre, Nijmegen, Netherlands; [2]Centre National de Recherche et de Formation sur le Paludisme, Ouagadougou, Burkina Faso; [3]Department of Plastic and Reconstructive Surgery, Azienda Ospedaliero Universitaria GiulianoIsontina Trieste, Trieste, Italy; [4]Department of Molecular Microbiology and Immunology, Johns HopkinsBloomberg School of Public Health, Baltimore, United States; [5]Liverpool School of Tropical Medicine, Liverpool, United Kingdom; [6]Department of Immunology and Infection, London School of Hygiene and Tropical Medicine, London, United Kingdom

**\*For correspondence:** teun.bousema@radboudumc.nl

†These authors contributed equally to this work

**Competing interest:** The authors declare that no competing interests exist.

**Sent for Review** 25 July 2023
**Preprint posted** 06 August 2023
**Reviewed preprint posted** 17 November 2023
**Reviewed preprint revised** 14 February 2024
**Version of Record published** 22 March 2024

**Abstract** It is currently unknown whether all *Plasmodium falciparum*-infected mosquitoes are equally infectious. We assessed sporogonic development using cultured gametocytes in the Netherlands and naturally circulating strains in Burkina Faso. We quantified the number of sporozoites expelled into artificial skin in relation to intact oocysts, ruptured oocysts, and residual salivary gland sporozoites. In laboratory conditions, higher total sporozoite burden was associated with shorter duration of sporogony (p<0.001). Overall, 53% (116/216) of infected *Anopheles stephensi* mosquitoes expelled sporozoites into artificial skin with a median of 136 expelled sporozoites (interquartile range [IQR], 34–501). There was a strong positive correlation between ruptured oocyst number and salivary gland sporozoite load ($\rho$ = 0.8; p<0.0001) and a weaker positive correlation between salivary gland sporozoite load and number of sporozoites expelled ($\rho$ = 0.35; p=0.0002). In Burkina Faso, *Anopheles coluzzii* mosquitoes were infected by natural gametocyte carriers. Among salivary gland sporozoite positive mosquitoes, 89% (33/37) expelled sporozoites with a median of 1035 expelled sporozoites (IQR, 171–2969). Again, we observed a strong correlation between ruptured oocyst number and salivary gland sporozoite load ($\rho$ = 0.9; p<0.0001) and a positive correlation between salivary gland sporozoite load and the number of sporozoites expelled ($\rho$ = 0.7; p<0.0001). Several mosquitoes expelled multiple parasite clones during probing. Whilst sporozoite expelling was regularly observed from mosquitoes with low infection burdens, our findings indicate that mosquito infection burden is positively associated with the number of expelled sporozoites. Future work is required to determine the direct implications of these findings for transmission potential.

## eLife assessment

This **important** study combines experimental infections with laboratory and field *Plasmodium falciparum* isolates to quantify the force of human malaria parasite transmission. By using **compelling** methodological approaches, the authors establish clear positive correlations between mosquito infection levels (as determined by oocyst numbers), sporozoite loads in salivary glands, and sporozoites expelled during feeding. The link between heterogeneous infection levels in the mosquitoes and malaria transmission would be of interest to vector biologists, parasitologists, immunologists, and mathematical modelers.

## Introduction

Malaria transmission to mosquitoes depends on the presence of mature gametocytes in human peripheral blood that are ingested by a mosquito during blood feeding. Ingested parasites undergo several developmental transformations in a process called sporogony. After ingestion, gametocytes transform into male and female gametes that fuse to form a zygote. The zygote differentiates into a motile ookinete that penetrates the midgut epithelium to form an oocyst. Multiple rounds of mitotic replication result in the formation of sporozoites inside an oocyst. Upon oocyst rupture, sporozoites are released into the haemocoel and invade the salivary glands (*Mueller et al., 2010*). These sporozoites penetrate the distal portion of the two lateral and medial lobes of the glands and accumulate extracellularly inside secretory cavities before entering the salivary ducts (*Wells and Andrew, 2019*). Despite the large number of sporozoites in the cavities, only a small proportion pass through the proximal part of the lobes where the salivary ducts become narrow (*Frischknecht et al., 2004*; *Bradley et al., 2018*), and only tens or low hundreds of sporozoites are assumed to be inoculated per mosquito bite (*Graumans et al., 2020*). Developmental bottlenecks during sporogony, as well as the size of the sporozoite inoculum, remain incompletely understood (*Graumans et al., 2020*).

The density of gametocytes in human peripheral blood is an important determinant of human to mosquito transmission. Though infections with parasite densities below the microscopic threshold for detection can infect mosquitoes (*Schneider et al., 2007*; *Bousema et al., 2006*), the likelihood and infection intensity increase with the number of ingested gametocytes (*Bradley et al., 2018*). Because of the abundance of low-density gametocyte carriers among infected populations, these are considered important drivers of malaria transmission (*Andolina et al., 2021*; *Slater et al., 2019*; *Ouédraogo et al., 2016*). Importantly, this conclusion is based on the assumption that all infected mosquitoes are equally infectious regardless of oocyst densities.

In apparent support of this assumption, single oocyst infections can result in thousands of salivary gland sporozoites (*Rosenberg and Rungsiwongse, 1991*; *Pringle, 1965*). A positive correlation between oocyst densities and salivary gland sporozoites was previously observed in *Plasmodium falciparum* (*Vaughan et al., 1992*; *Miura et al., 2019*) as well as in *Plasmodium vivax* (*Solarte et al., 2011*), suggesting that on average of 1000–2000 sporozoites reach the salivary glands per single ruptured oocyst. The few studies that quantified sporozoite inoculum by allowing mosquitoes to salivating into capillary tubes containing mineral oil, sucrose solution, or blood (*Beier et al., 1991a*; *Beier et al., 1991b*; *Rosenberg et al., 1990*) estimated median inocula ranging between 8 and 39 sporozoites with a minority of mosquitoes expelling >100 sporozoites (reviewed in *Graumans et al., 2020*).

While these studies provide some insights into sporozoite expelling and transmission dynamics, they do not reflect natural feeding conditions. Also, microscopy techniques used to quantify sporozoites may have underestimated the number of sporozoites (*Medica and Sinnis, 2005*). Subsequent studies with rodent *Plasmodium* models allowed natural mosquito probing through skin and showed a positive association between sporozoite density in salivary glands and expelled sporozoite numbers (*Medica and Sinnis, 2005*; *Aleshnick et al., 2020*). A recent study using a *Plasmodium yoelii* malaria model demonstrated that mosquitoes with over 10,000 salivary gland sporozoites were 7.5 times more likely to initiate a malaria infection in mice (*Aleshnick et al., 2020*). This finding has not been replicated for human malarias but is broadly consistent with post hoc analysis of infection likelihood in controlled human malaria infections (CHMI), where only mosquitoes that had >1000 *P. falciparum* salivary gland sporozoites remaining after probing (the residual sporozoite load) were capable of establishing an infection in malaria-naïve volunteers (*Churcher et al., 2017*).

If low oocyst/low sporozoite densities in mosquitoes are unlikely to initiate infections in humans, this may have profound consequences for our understanding of transmission (*WHO, 2017*). If mosquitoes

with low-infection burdens have limited transmission potential, then the rationale for targeting low-density infections in humans that give rise to low-infection burdens in mosquitoes (*Bradley et al., 2018*) may be diminished.

Here, we examined the progression of sporozoite development and the number of sporozoites expelled into artificial skin by individual *Anopheles* mosquitoes infected either with *P. falciparum* gametocyte cultures or in experimental infections with naturally circulating parasite strains. We directly assessed the association between oocyst burden, salivary gland infection intensity, and the number of sporozoites expelled.

## Results

### Low number of *P. falciparum* sporozoites are quantifiable by qPCR

Multicopy mitochondrial COX-1 and 18S rRNA gene targets were analyzed in octuplicate on serial dilutions of sporozoites to assess qPCR performance and select the target that achieved highest sensitivity and most consistent sporozoite detection. COX-1 outperformed 18S in detecting sporozoites (*Figure 1—figure supplement 1*), the limit of detection (LOD) and limit of quantification (LOQ) for

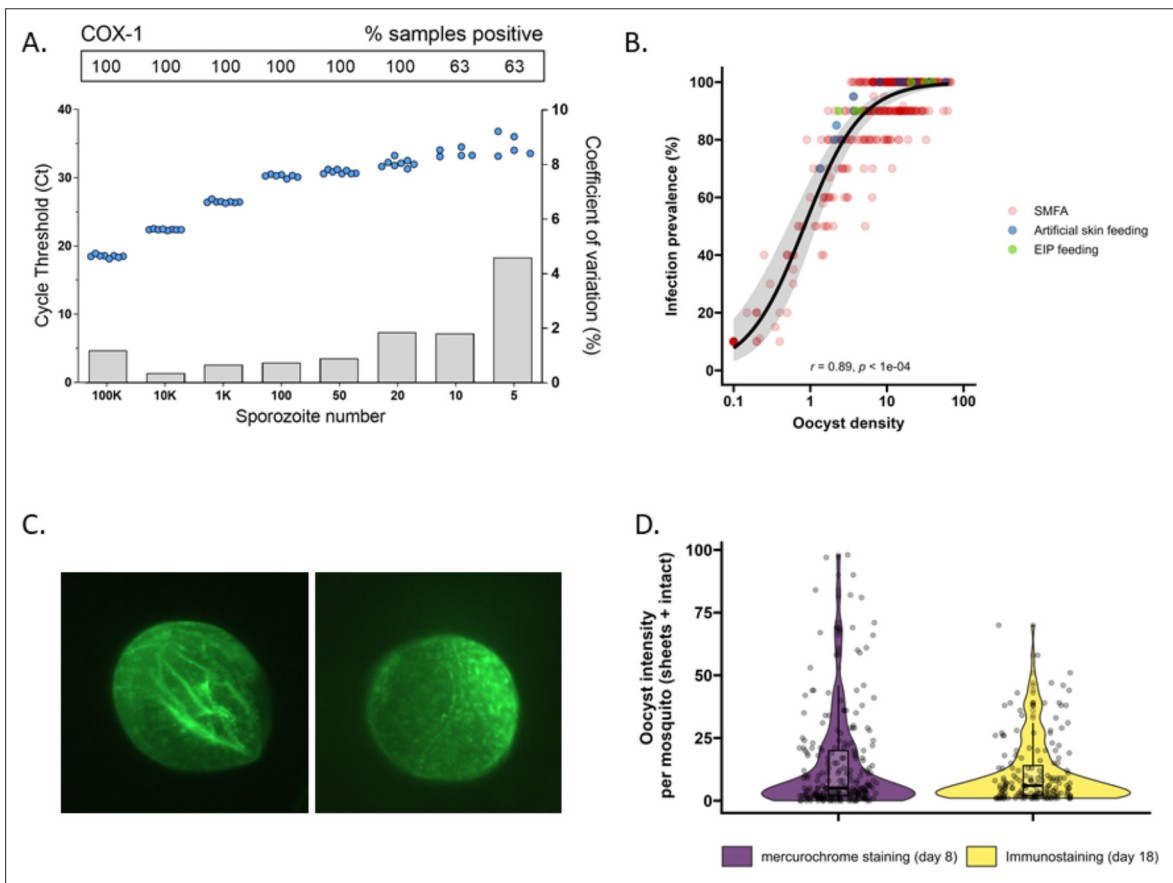

**Figure 1.** The detectability of sporozoites by molecular methods and oocysts by immunolabeling. (**A**) qPCR performance for *P. falciparum* sporozoites. Serial dilutions of sporozoites (x-axis) were prepared in phosphate-buffered saline (PBS) and assessed in octuplicate on a single plate to determine qPCR COX-1 limit of detection and quantification. Dots represent sample cycle threshold (left y-axis), and bars coefficient of variation (right y-axis). For each serial dilution, Ct sample positivity is shown as the percentage of total tested. (**B**) The relationship between oocyst density versus infection prevalence from 457 membrane feeding experiments using cultured gametocytes. Colors represent regular feeds (red) and those selected for performing experiments on the extrinsic incubation period (EIP; green) or sporozoite-expelling experiments (blue). (**C**) Immunofluorescence staining with 3SP2-Alexa 488 anti-CSP. Empty sheet (left) and intact oocyst (right). (**D**) Violin plots of oocyst staining – day 8 post infection by mercurochrome (purple) and – day 18 by 3SP2-Alexa 488 anti-CSP immunostaining (yellow) for cultured parasites. Box plots show interquartile range, whiskers show the 95% intervals.

The online version of this article includes the following figure supplement(s) for figure 1:

**Figure supplement 1.** qPCR performance on extracted *P. falciparum* sporozoites (SPZ) targeting the 18S rRNA gene, and DNA extraction/quantification of *P. falciparum* SPZ from artificial skin.

COX-1 qPCR were determined at 20 sporozoites per sample (8/8 sample positivity with a coefficient of variation <2) (*Figure 1A*). Next, we confirmed the qPCR performance in combination with the matrix that was used for expelling experiments by spotting serial dilutions of sporozoites in whole-blood on Integra dermal substitute artificial skin (*Agostinis et al., 2021*) prior to nucleic acid extraction. The matrix had no apparent impact on sporozoite detectability and quantification (*Figure 1—figure supplement 1*), unlike previously used filter paper matrices (*Brugman et al., 2018*; *Holzschuh and Koepfli, 2022*).

## A comparative analysis of oocyst densities using mercurochrome staining and immunostaining

Subsequently, mosquito feeding assays were performed by offering diluted in vitro cultured gametocytes to mosquitoes to obtain a broad range of oocyst densities. The association between $\log_{10}$ oocyst intensity and infection prevalence in mosquitoes was assessed using a logistic regression model (using data from 457 mosquitoes, *Figure 1B*). Mosquito infection prevalence was strongly associated with oocyst intensity, corroborating earlier work (*Churcher et al., 2012*), with a strong positive sigmoidal association and a 14.68 (95% CI, 8.18–26.35, p<0.0001) times higher odds of infection prevalence associated with a tenfold higher oocyst density. In this analysis, oocysts were enumerated microscopically following standard mercurochrome staining. We previously used 3SP2-Alexa 488 anti-circumsporozoite (CSP) immunostaining to visualize ruptured and intact oocysts (*Stone et al., 2013*). The concordance between oocyst prevalence by standard oocyst mercurochrome staining (day 8 post infection [PI]) and anti-CSP immunostaining on day 18 PI (*Figure 1C*) was investigated. For this, oocyst density distributions by both methods were compared within batches of mosquitoes that were fed on cultured gametocytes during the same standard membrane feeding assay (*Figure 1D*). We observed no statistically significant difference in oocyst densities determined by day 8 mercurochrome staining (median five oocysts, interquartile range [IQR], 2–20, N = 252) and day 18 immunolabeling (median six oocysts, IQR, 2–14, N = 167; Student's *t*-test on log densities, p=0.944).

## Highly infected mosquitoes become salivary gland sporozoite positive earlier

Following assay validation, the extrinsic incubation period (EIP) was compared between mosquitoes with low and high oocyst densities. Batches of high- and low-infected mosquitoes were generated using standard membrane feeding assay with standard concentrations of cultured gametocytes or culture material that was five- or tenfold diluted (*Figure 1B*). On day 8 PI, 20 mosquitoes were dissected and batches that had ≥70% oocyst infection prevalence and means of ≤5 or >20 oocysts were selected for subsequent dissections (*Figure 1B*). On days 9–11, salivary glands and the remaining mosquito body

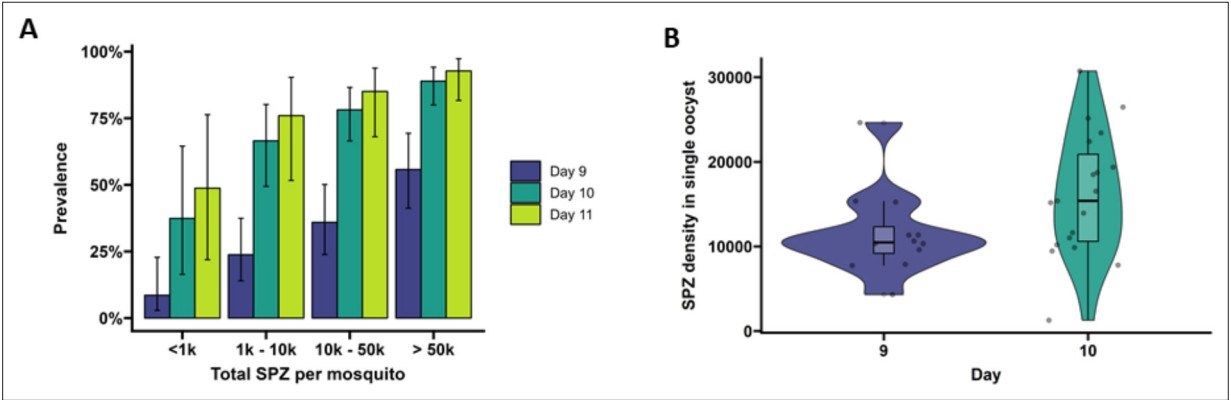

**Figure 2.** Extrinsic incubation period in high- versus low-infected mosquitoes. (**A**) Total sporozoites (SPZ) per mosquito in body plus salivary glands (x-axis) were binned by infection load <1k; 1k–10k; 10k–50k; >50k and plotted against the proportion of mosquitoes (%) that were SPZ positive (y-axis) as estimated from an additive logistic regression model with factors day and SPZ categories. In total, 120, 120, and 40 mosquitoes were dissected on days 9 (blue), 10 (dark green), and 11 (light green), respectively. Error bars show the 95% confidence intervals. (**B**) Violin plots of SPZ density in single oocysts dissected on days 9 (purple) and 10 (green). The box indicates the interquartile range (IQR) (Q1 and Q3 quartiles) and the median. Lines extending Q1 and Q3 quartiles indicate the range of the data within 1.5× IQR.

(that included the mosquito midgut) were collected separately and analyzed for sporozoite density by COX-1 qPCR. Mosquitoes were then binned into four categories of sporozoite infection intensity, defined as the sum of mosquito body and salivary gland sporozoite density (*Figure 2A*). This total sporozoite density was examined in relation with the likelihood of being salivary gland sporozoite positive and thus having completed sporogonic development.

On day 9 PI, 54.3% of highly infected mosquitoes (>50,000 sporozoites) were salivary glands sporozoite positive (*Figure 2A*) and had 3.17 (95% CI 95%, 0.7–14.4, p=0.4278) times the odds of being salivary gland positive compared to low-infected mosquitoes with <1000 sporozoites (27.3% salivary gland sporozoite positive). By day 10 PI, 82.2% of mosquitoes with 10,000–50,000 (10k–50k) sporozoites were salivary glands positive and had 11.56 (95% CI, 1.83, 73.25, p=0.0449) times the odds of being salivary glands positive compared to low-infected mosquitoes with <1000 sporozoites (28.6% salivary glands sporozoite positive). On day 11 PI, all 15 highly infected mosquitoes (>50,000 sporozoites) were salivary glands sporozoite positive and meaningful odds ratios and 95% CIs could not be determined. When considering the entire period over which EIP experiments were conducted, mosquitoes with >50,000 sporozoites had a 13.44 times higher odds of being salivary glands positive compared to low-infected mosquitoes (<1000 sporozoites; 95% CI, 4.02–44.88, p<0.0001). Mosquitoes harboring 10,000–50,000 sporozoites had a 5.98 times higher odds of being salivary glands positive compared to low-infected mosquitoes (95% CI, 1.88–19.07, p=0.0119). These data demonstrate that EIP is shorter in high-infected compared to low-infected mosquitoes in a temperature- and humidity-controlled environment.

## Sporozoite densities increase with oocyst age

To quantify the number of sporozoites per oocyst, individual oocysts were isolated from midguts on days 9 and 10 PI (*Soontarawirat et al., 2017*) and stained with 1% mercurochrome. The median sporozoite density was 10,485 (IQR, 9171.3–12,322.5; 12 examined mosquitoes) per oocyst for day 9 and 15,390 (IQR, 10,600–20,887, 19 examined mosquitoes) for day 10 (*Figure 2B*, p=0.04995, by Welch's two-sample *t*-test on $\log_{10}$-transformed data).

## Oocyst density, salivary gland density, and the size of the sporozoite inoculum are positively associated in mosquitoes infected with cultured gametocytes

We performed artificial skin feeding experiments with individual mosquitoes on day 15 PI to assess sporozoite expelling. To avoid interference of residual blood with oocyst immunolabeling, we assessed mosquito oocyst density (ruptured and intact oocysts) and sporozoite density in the salivary glands on day 18, allowing 3 d for bloodmeal digestion. This approach allowed us to determine the density of intact and ruptured oocysts and associate this to sporozoite density in the same mosquito. It was noted that a minority of oocysts failed to rupture during this time span; 5% (93/1854) of all oocysts were visually intact and 88.3% (166/188) of examined mosquitoes had at least one unruptured oocyst on day 18. While we observed good concordance between oocyst densities by mercurochrome staining on day 8 and immunostaining on day 18 PI (*Figure 1D*), oocysts sporadically did not take up the 3SP2-Alexa 488 anti-CSP antibody labeling. In 54% (12/22) of mosquitoes without evidence of ruptured oocysts, we observed salivary gland sporozoites. Nevertheless, there was a strong positive association observed between ruptured oocysts and salivary gland sporozoite load ($\rho$ = 0.80, p<0.0001; N = 185) (*Figure 3B*). When intact oocysts were also included, this association was nearly identical ($\rho$ = 0.80, p<0.0001; N = 185). We estimated a median of 4951 (IQR, 3016–8318) salivary gland sporozoites per ruptured oocyst in *Anopheles stephensi*. Next, the association between sporozoite density and the number of expelled sporozoites (inoculum size) was determined. For this, individual mosquitoes in miniature cages were allowed to probe for a maximum of 8 min on blood-soaked artificial skin. In preparation of expelling experiments, we assessed if sporozoites may migrate away from the examined area of artificial skin (surface feeding area 201 mm²) using a batch of high-infected mosquitoes (35,6 oocyst mean). For this, we examined the part of the artificial skin normally unexposed to mosquitoes beneath the rubber band. In three experiments with either 5 (N = 2) or 20 mosquitoes (N = 1) probing and feeding on the artificial skin, we detected a total of 3264, 3599, and 10,340 expelled sporozoites in the exposed skin part, respectively, but no qPCR signal in the

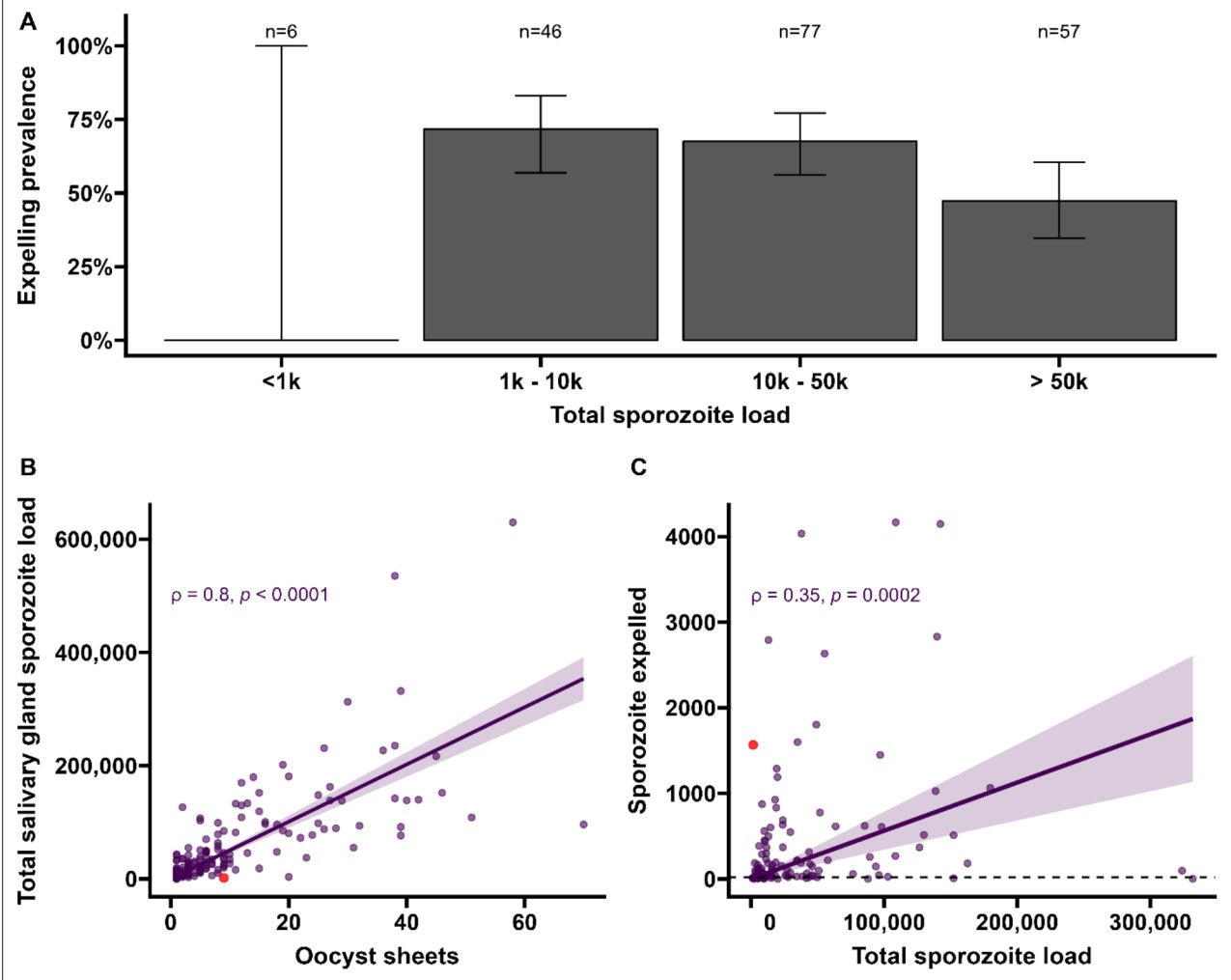

**Figure 3.** Sporozoite expelling in relation to infection burden in *Anopheles stephensi* mosquitoes infected with cultured gametocytes. (**A**) Binning of mosquitoes by total sporozoite load and expelling prevalence (N = 186). (**B**) The number of ruptured oocysts stained by 3SP2-Alexa 488 anti-CSP and fluorescent microscopy (x-axis) in relation to total salivary gland sporozoite density (y-axis), assessed by COX-1 qPCR; $\rho$ = 0.80 (CI, 0.74–0.85, p<0.0001). The red dot indicates a mosquito which had 9 ruptured oocysts but only 126 residual salivary glands sporozoites while it expelled 1567 sporozoites. Considering the high number of ruptured oocysts in the midgut, it is possible that some lobes of salivary glands were missed during dissection and sporozoite load was underestimated by qPCR. (**C**) Total sporozoite density (residual salivary gland sporozoites + sporozoites expelled, x-axis) in relation to the number of expelled sporozoites (y-axis) by COX-1 qPCR $\rho$ = 0.35 (CI, 0.17–0.50, p=0.0002). The dotted line on the x-axis shows the threshold of qPCR detection of 20 sporozoites. The line represents the fitted linear regression line, the intercept is forced to zero for biological plausibility, and the gray shaded area is the 95% CI.

The online version of this article includes the following source data and figure supplement(s) for figure 3:

**Source data 1.** Source data include re-analyses of correlation coefficients when different thresholds for data inclusion are imposed to determine whether associations are robust.

**Figure supplement 1.** Sporozoite expelling in relation to infection burden.

unexposed part of artificial skin. This indicates no or negligible migration of sporozoites to the outer sides of the skin and affirms that our procedure reliably captures all expelled sporozoites.

Among all mosquitoes used in skin feeding experiments, 53% (116/216) expelled sporozoites at any density, and 45% (97/216) expelled sporozoites above our threshold for reliable detection and quantification of 20 sporozoites/skin. In line with previous work with rodent malaria species *Plasmodium berghei* (*Matsuoka et al., 2002*), sporozoite expelling was observed in mosquitoes that did not take a bloodmeal; 33% (5/15) of mosquitoes that probed but failed to take a bloodmeal expelled sporozoites (sporozoite range 5–1802). To examine sporozoite expelling in relation to infection burden, mosquitoes were binned into four categories of salivary gland infection intensity that

was estimated by combining the residual sporozoite load in the salivary glands and the sporozoites successfully expelled into the skin. In this way also heavily infected mosquitoes that expelled the majority of sporozoites were categorized as heavily infected. We observed a strong positive association between oocyst sheets and total salivary gland sporozoite load (Spearman's correlation coefficient $\rho$ = 0.80, p<0.0001; N = 111; *Figure 3B*). When examining this association for different ranges of oocyst intensity (<5, <10, <20 oocysts), correlation estimates remained highly similar and statistical significance was retained (*Figure 3—figure supplement 1*).

We observed no statistically significant association between salivary gland infection intensity and the prevalence of expelling sporozoites (*Figure 3A*; 95% CI, 0.74–0.85; p=0.1880). Among mosquitoes that expelled sporozoites, the medians of expelled and residual salivary gland sporozoites were 136 (IQR, 34–501) and 23,947 (IQR, 9127–78,380), respectively, while the highest number of sporozoites detected in skin was 4166. We observed a weak but statistically significant positive association between total sporozoite load and the number of expelled sporozoites ($\rho$ = 0.35, 95% CI, 0.17–0.50; p=0.0002; N = 112; *Figure 3C*). When examining this association for different ranges of total sporozoite load (<10,000; <50,000; <100,000 sporozoites), correlation estimates remained highly similar although this correlation lost statistical significance when only including low total sporozoite loads <10,000 sporozoites ($\rho$ = 0.29; 95% CI, –0.07 to 0.58, p=0.1094) (*Figure 3—source data 1*). When we included 26 observations from mosquitoes that did not expel any sporozoites, we observed no statistically significant association between total sporozoite load and the number of expelled sporozoites ($\rho$ = 0.016, 95% CI, –0.12 to 0.16; p=0.8321).

We observed no evidence for a sharp increase in sporozoite expelling at sporozoite densities ≥10,000, as was previously described in rodent malaria models *Aleshnick et al., 2020*; 28% (53/186) of our mosquitoes harbored sporozoites below this density. Among these low-infected mosquitoes, 64% (34/53) expelled sporozoites and the median number of expelled sporozoites was 67 (IQR, 13–128).

## Infected mosquitoes in Burkina Faso show comparable correlations between oocyst density, salivary gland density, and sporozoite inoculum

Seven gametocyte donors (age 5–15 y; median 48 gametocytes/µl [range 40–167]) were recruited in Balonghin, Burkina Faso. Their blood was offered to locally reared *Anopheles coluzzii* via membrane feeders at the gametocyte density observed and following gametocyte enrichment by magnetic activated cell-sorting whereby gametocyte concentration was increased by approximately three- to fourfold. Five donors infected mosquitoes in at least one of these two conditions (*Figure 4A*); as expected, mosquito infection rates and oocyst densities were significantly increased after gametocyte enrichment. Mosquito batches with ≥50% infection prevalence were used for artificial skin feeding experiments as described above. From a total of 53 mosquitoes, salivary gland sporozoites were detected in 69% (37/53). Also, 31 mosquito midguts were available for immunolabeling, of which 67.7% (23/31) had oocyst sheets detected by immunolabeling on day 19 and nearly half of these (48% [15/31]) had <5 oocyst sheets. Two mosquito midguts were negative by immunolabeling while their residual salivary gland sporozoite loads were 7958 and 14,750 respectively, suggesting oocyst staining failure or rupture followed by complete oocyst disappearance. 87.8% (370/421) of all detected oocysts ruptured while 39% (9/23) of oocyst positive mosquitoes harbored at least one intact oocyst (range 1–19). Failure to rupture was uncommon in low-infected mosquitoes (≤5 oocysts, N = 80) where only two midguts had one intact oocyst. Among 53 mosquitoes used for artificial skin feeding on day 16 PI, 89% (33/37) salivary gland sporozoite-positive mosquitoes with a median of 45,100 residual salivary gland sporozoites (IQR, 20,310–164,900) expelled a median of 1035 sporozoites (IQR, 171–2969). We estimated a median of 6350 (IQR, 4225–8475) salivary gland sporozoites per ruptured oocyst in these experiments. Three skin samples were positive by COX-1 qPCR (range 1–64) while salivary glands were negative, suggesting either all sporozoites were expelled or a technical failure in DNA extraction from the salivary glands. AMA-1 amplicon sequencing was conducted on extracted salivary glands and artificial skins for mosquitoes that expelled sporozoites at any quantity. Among 22 mosquitoes that were infected from three gametocyte donors, we identified 10 unique clones; 68.2% (15/22) of the infected mosquitoes that were tested harbored multiclonal infections. Following probing and successful sporozoite expelling, 10 skin samples contained more than one *P. falciparum* clone (45.5%; 10/22) (*Figure 5*).

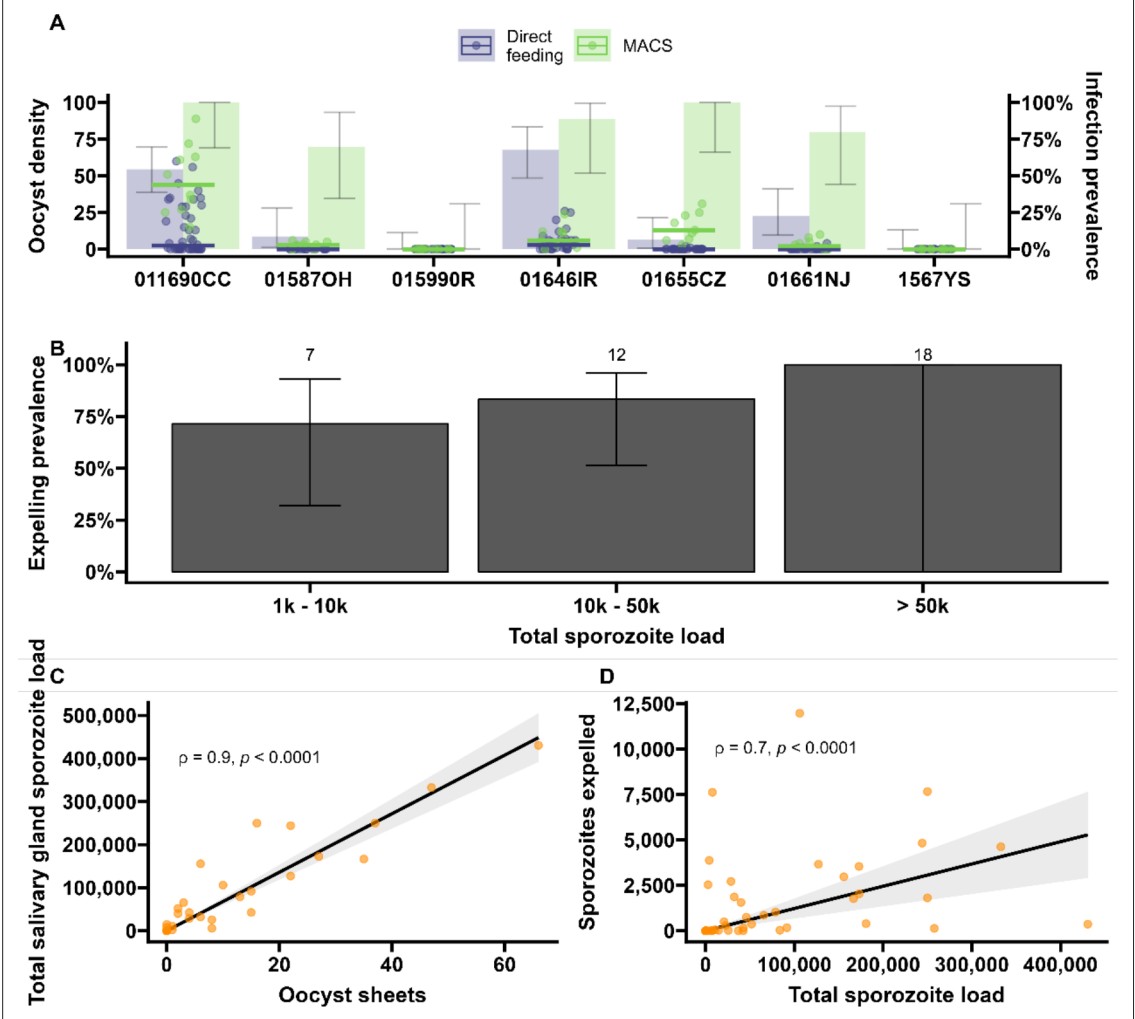

**Figure 4.** Sporozoite expelling in relation to infection load in *Anopheles coluzzii* mosquitoes infected by naturally acquired gametocyte infections in Burkina Faso. (**A**) Direct feeding (blue) vs magnetic-activated cell sorting (MACS; green). Bars show the infection prevalence for each of the seven gametocyte carriers. Scatter plots with median lines show the midgut oocyst density as a result of direct feeding (blue) and MACS (green). (**B**) Binning of total sporozoite load and expelling prevalence (N = 25). (**C**) Scatter plot of absolute numbers of ruptured oocyst (sheet) density assessed by fluorescent microscopy vs total salivary gland sporozoite density assessed by COX-I qPCR; $\rho$ = 0.90 (95% CI, 0.80–0.95). The line represents the fitted linear regression line and the gray shaded area is the 95% CI. (**D**) Scatter plot of absolute numbers of total sporozoite density (residual salivary gland sporozoites + sporozoites expelled) and sporozoites expelled into the artificial skin assessed by COX-I qPCR; $\rho$ = 0.70 (CI, 0.52–0.82). The line represents the fitted linear regression line, the intercept is forced to zero for biological plausibility, and the gray shaded area is the 95% CI.

The online version of this article includes the following source data for figure 4:

**Source data 1.** Source data include re-analyses of correlation coefficients when different thresholds for data inclusion are imposed to determine whether associations are robust.

During the artificial skin feeding, 30% (16/53) of probing mosquitoes did not ingest blood, of which 68% (11/16) expelled sporozoites (range 1–11,970). There appeared to be a trend toward higher prevalence of expelling with increasing sporozoite density (*Figure 4B*). There was a strong association between ruptured oocyst density and total salivary gland sporozoite density ($\rho$ = 0.84, 95% CI, 0.80–0.95; p<0.0001) (*Figure 4C*); when intact oocysts were also included, the association was very similar ($\rho$ = 0.86, p<0.0001; N = 30). When examining these associations for different ranges of oocyst intensity (<5, <10, <20 oocysts), correlation estimates remained highly similar and statistical significance was retained (*Figure 4—source data 1*). There was also a strong positive association between total sporozoite load and the number of sporozoites expelled ($\rho$ = 0.71, 95% CI, 0.52–0.82; p<0.0001) (*Figure 4D*). When examining this association between total sporozoite load and expelling

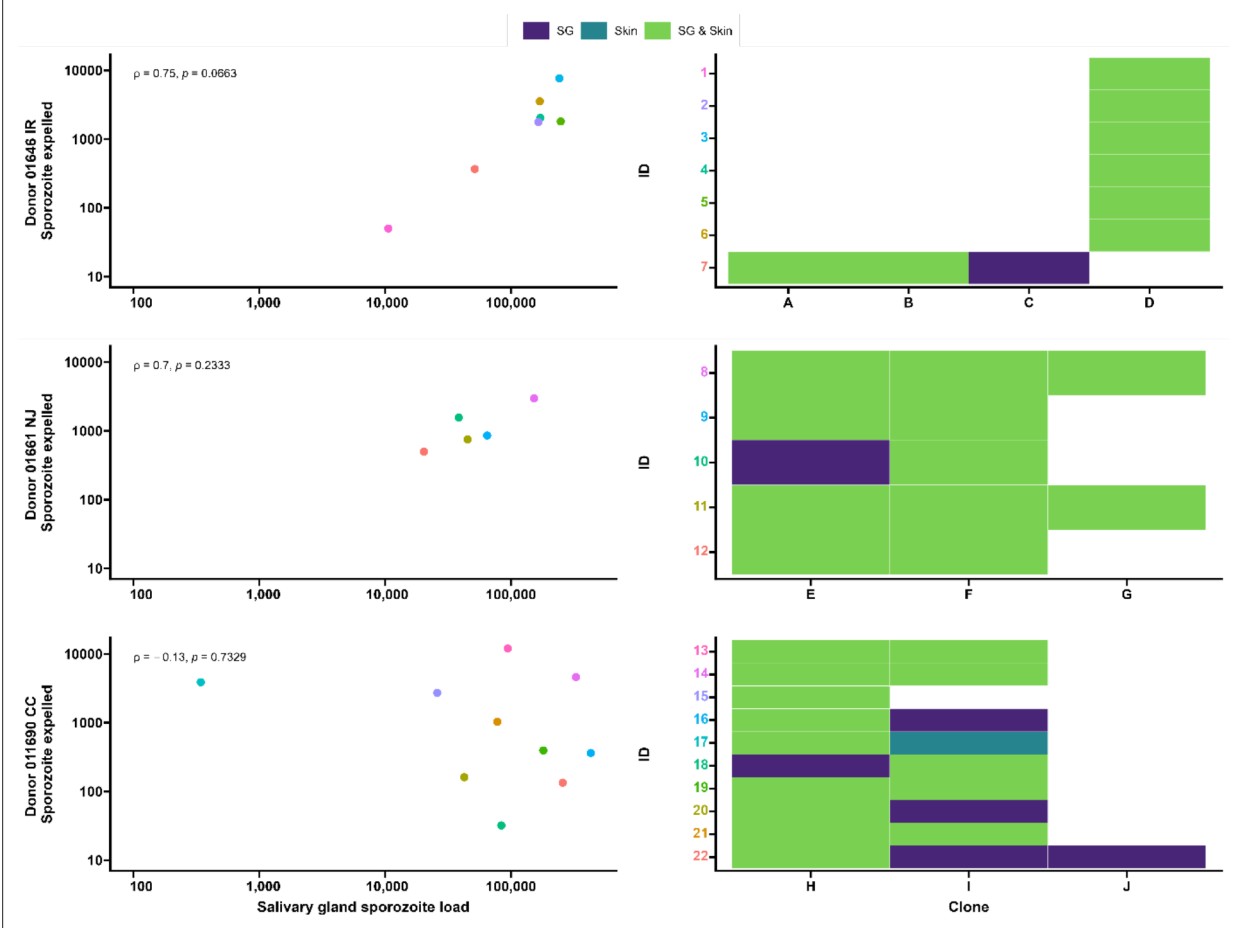

**Figure 5.** Clonal complexity of *P. falciparum* infections in salivary glands and artificial skins following probing by mosquitoes infected by gametocyte carriers who were naturally infected in Burkina Faso. Clonal data for three donors: 01646 IR (top row), 01661 NJ (middle row), and 011690 CC (bottom row). Left panel shows scatter plots for the association between sporozoite salivary gland load and expelled sporozoites in skin, with a Spearman's rank correlation across samples from each donor. Right panel shows clonal data for each donor sample in a heatmap plot. The colored numbers on y-axes correspond to the color of the sample in the scatter plot. Purple indicates the presence of a clone in the salivary gland only, turquoise indicates the presence of a clone in the skin only, and green indicates the presence of a clone in both salivary gland and skin.

for different ranges of total sporozoite load (<10,000, <50,000, <100,000 sporozoites), correlation estimates remained similar and statistically significant (*Figure 4—source data 1*).

## Discussion

We examined *P. falciparum* sporogony in different mosquito species by employing sensitive molecular and staining techniques in conjunction with mosquito probing experiments on artificial skin. Through the visualization of ruptured oocysts and the simultaneous quantification of sporozoites that are expelled by individual mosquitoes, we observed that 95% of oocysts rupture to release sporozoites and that higher salivary gland sporozoite load is associated with shorter time to colonization of salivary glands and larger inoculum size.

The proportion of infectious mosquitoes is a central component of malariometric indices both in terms of quantifying the force of infection and the human infectious reservoir. The entomological inoculation rate (EIR) is defined as the number of infectious bites per person per time unit and is the product of human biting rate and the proportion of sporozoite-positive mosquitoes (*Tusting et al., 2014*; *Shaukat et al., 2010*). While EIR is a common measure of human malaria exposure, mosquito infection prevalence is used in this calculation and thus assumes all sporozoite-positive mosquitoes are equally infectious (*Smith et al., 2012*). Similarly, assessments of the human infectious reservoir for malaria typically take the number of oocyst-positive mosquitoes as a measure of transmission

(*Ouédraogo et al., 2016*; *Gonçalves et al., 2017*) and thereby not only assume that all oocysts will lead to salivary gland sporozoites but also that all oocyst-positive mosquitoes have equal transmission potential. Recent work with a rodent malaria model challenged this central assumption (*Aleshnick et al., 2020*); we provide the first direct evidence for *P. falciparum* that sporozoite burden may indeed be a relevant determinant in efficient sporozoite expelling.

In this study, we assessed sporozoite expelling by mosquitoes carrying low- and high-infection burdens. Our findings confirm that the vast majority (~95%) of oocysts rupture to release sporozoites. This estimate is higher than a previous study with cultured gametocytes (~72%) (*Stone et al., 2013*) that did not provide a second bloodmeal that may accelerate oocyst maturation (*Shaw et al., 2020*); a second bloodmeal also better mimics natural feeding habits where multiple bloodmeals are taken within the period required for sporogony. Moreover, we observed a strong positive association between sporozoite salivary gland load and ruptured oocyst density in mosquitoes infected with both cultured and naturally circulating gametocytes, with similar median numbers of 4951 (IQR, 3016–8318) and 6350 (IQR, 4225–8475) salivary gland sporozoites per ruptured oocyst in *An. stephensi* and *An. coluzzii,* respectively. Previous studies have shown that despite the substantial release of sporozoites per oocyst, only a proportion of sporozoites successfully reach the salivary glands (*Rosenberg and Rungsiwongse, 1991*; *Hillyer et al., 2007*). In line with this, we observed considerably higher sporozoite estimates in intact oocysts (10,485–15,390 sporozoites per oocyst). Whilst these sporozoite estimates are higher than commonly reported (*Graumans et al., 2020*), an early study from the 1960s reported microscopy-detected sporozoite densities above 9000 sporozoites per oocyst (*Pringle, 1965*) and a recent study using qPCR similarly observed up to 12,583 sporozoites from a single oocyst (*Wang et al., 2018*). These high sporozoite numbers per oocyst also mean that, different from *P. yoelii* (*Aleshnick et al., 2020*) and *P. berghei* (*Churcher et al., 2017*), only a minority of infected mosquitoes had sporozoite densities below the threshold values previously reported in relation to a very low likelihood of achieving secondary infections. Even among infected *An. stephensi* mosquitoes with only one ruptured oocyst, 92.9% (26/28) had >1000 and 35.7% (10/28) had >10,000 sporozoites in their salivary glands. In our experiments in natural gametocyte carriers in Burkina Faso, only two mosquitoes were observed with single oocysts and both had >10,000 salivary gland sporozoites. Although *P. falciparum* sporozoite densities below these lower thresholds were thus uncommon, we observed that 39% of sporozoite-positive mosquitoes across a wide range of infection densities failed to expel sporozoites upon probing. This finding broadly aligns with an earlier study of Medica and Sinnis that reported that 22% of *P. yoelii*-infected mosquitoes failed to expel sporozoites (*Medica and Sinnis, 2005*). For highly infected mosquitoes, this inefficient expelling has been related to a decrease in apyrase in the mosquito saliva (*Rossignol et al., 1984*; *Thiévent et al., 2019*).

Importantly, we observed a positive association between salivary gland sporozoite density and the number of expelled sporozoites. For unknown reasons, this association was markedly stronger in experiments where *An. coluzzii* mosquitoes were infected using blood from gametocyte carriers who were naturally infected with *P. falciparum* in Burkina Faso compared to *An. stephensi* mosquitoes infected with cultured gametocytes. In our experiments with natural gametocyte carriers, sporozoite density appeared associated with both the likelihood of expelling any sporozoites and the inoculum size. The most heavily infected mosquitoes expelled ~14-fold higher sporozoite numbers compared to the lowest quartile. In addition, we found evidence that the EIP – the period required for a mosquito to become salivary gland sporozoite positive – is shorter for heavily infected mosquitoes. Previous studies have not found consistent effects of parasite burden on EIP (*Guissou et al., 2023*; *Stopard et al., 2021*) but, unlike our experiments, were also not specifically designed to examine this association across a broad range of infection intensities. Our associations of a shorter EIP in highly infected mosquitoes and, in separate experiments, a larger sporozoite inoculum size for highly infected mosquitoes make mosquito infection intensity a plausible factor in determining onward transmission potential to humans. Heavily infected mosquitoes may be infectious sooner and be more infectious. On the other side of the infection spectrum, it is conceivable that submicroscopic gametocyte carriers that typically result in low oocyst burdens in mosquitoes give rise to infected mosquitoes with a reduced transmission potential. This would greatly reduce their importance for sustaining malaria transmission and make it less important to identify individuals with low parasite densities for malaria elimination purposes (*Slater et al., 2015*). At the same time, we occasionally observed high sporozoite inocula from mosquitoes with low infection intensity. This argues against a clear threshold

sporozoite or oocyst density below which mosquitoes are truly irrelevant for transmission. Moreover, the high number of sporozoites per oocyst makes it plausible that even the low oocyst burdens that are typically observed from asymptomatic parasite carriers (in the range of 1–5 oocysts/infected gut; *Andolina et al., 2021*; *Gonçalves et al., 2017*) would be sufficient to result in salivary gland sporozoite loads that are sufficient to result in secondary infections.

Our study leaves a number of questions and has several limitations. While the use of two mosquito and gametocyte sources was a relevant strength of our study, an uncertainty relates to the choice of artificial skin that has a realistic 1.33 mm thickness but is arguably less natural than microvascularized skin with all the natural cues for mosquito probing. Whilst genuine skin might have improved natural feeding behavior, probing and blood feeding were highly efficient in our model and we see no reasons to assume bias in the comparison between high- and low-infected mosquitoes. Our assessments of EIP and sporozoite expelling did not demonstrate the viability of sporozoites. Whilst the infectivity of sporozoites at different time points PI has been examined previously (*Yang et al., 2017*), these experiments have never been conducted with individual mosquitoes. In vitro experiments that aim to determine the infectivity of single mosquito bites would ideally retain the skin barrier that may be a relevant determinant for invasion capacity and use primary hepatocytes. These experiments were beyond the scope of the current work and would also not provide conclusive evidence on the likelihood of achieving secondary infections. Given striking differences in sporozoite burden between different *Plasmodium* species – low sporozoite densities appear considerably more common in mosquitoes infected with *P. yoelii* and *P. berghei* (*Graumans et al., 2020*; *Churcher et al., 2017*; *Ahmed et al., 2021*), and the inherent limitations of in vitro studies – the association between sporozoite inoculum size and the likelihood of achieving secondary infections may be best examined in controlled human infection studies. CHMI experiments can be specifically designed to estimate the likelihood that probing by (individual) high- and low-infected mosquitoes results in blood-stage infection in malaria-naïve volunteers. Whilst laborious, CHMI studies are unique in allowing definitive evidence on the possible differences in infectiousness between high- and low-infected mosquitoes.

In conclusion, we observed that the majority of oocysts rupture and contribute to salivary gland infection load. We further observe that this sporozoite load is highly variable and an important determinant of the number of sporozoites that is expelled into the skin upon probing.

## Materials and methods

### *P. falciparum* in vitro culture and mosquito infection

*P. falciparum* gametocytes, NF54 (West Africa) and NF135 (Cambodia) (*Conway et al., 2000*; *Teirlinck et al., 2013*) were cultured in an automated culture system (*Ponnudurai et al., 1982*) and maintained as previously described at Radboudumc, Nijmegen (*van de Vegte-Bolmer et al., 2021*). *An. stephensi* mosquitoes, Nijmegen Sind-Kasur strain (*Feldmann and Ponnudurai, 1989*), were reared at 30°C and 70–80% humidity with a 12 hr reverse day/night cycle. To have a range of infection intensities in mosquitoes, undiluted and diluted cultured gametocytes (0.3–0.5% gametocytes) were generated in heparin blood. In total, 100–150, 1- to 3-day-old female mosquitoes were fed using glass membrane mini-feeders (*Ponnudurai et al., 1989*).

### Mosquito feeding on gametocyte carriers who were naturally infected with *P. falciparum*

*P. falciparum* gametocyte donors were recruited at schools in Saponé Health District, 45 km southwest of Ouagadougou. Following informed consent, finger-prick blood was examined for gametocytes by counting against 500–1000 white blood cells (WBCs) in thick-blood films. The gametocyte counts were done by two independent microscopists and expressed as density/µl by assuming 8000 WBCs per µl of blood. If gametocyte densities were above 16/µl, 2–5 ml of venous blood was drawn by venepuncture in lithium heparin tubes (BD Vacutainers, ref 368496) and transported to the Centre National de Recherche et de Formation sur le Paludisme (CNRFP) insectary in Ouagadougou in thermos flasks filled with water at 35.5°C (*Soumare et al., 2021*).

Then, 700 µl of whole blood was used for immediate feeding, which was performed as described elsewhere, using 3- to 5-day-old *An. coluzzii* mosquitoes per glass mini feeder (Coelen Glastechniek, the Netherlands) that was attached to a circulating water bath set up at 38°C (Isotemp, Fisher

Scientific) (*Ouédraogo, 2013*; *Musiime et al., 2019*). Fifty mosquitoes per cup (starved for 12 hr) were allowed to feed in the dark for 15–20 min through a Parafilm membrane. To increase mosquito infection prevalence and intensity, 1 ml of blood was used to enrich gametocytes with magnetic cells sorting columns (MACS), as described by *Graumans et al., 2019*. In this process, a prewarmed 23G hypodermic needle (Becton Dickinson, Franklin Lakes, NJ) was attached to an LC column on a QuadroMACS separator (Miltenyi Biotec, UK) that was placed inside a temperature-monitored incubator (37°C). The flow-through was collected in a 15 ml Falcon tube. Following hydration with 1 ml RPMI, 1 ml donor blood (lithium heparin) was added to MACS column. The column was rinsed with 2 ml warm RPMI. The 15 ml Falcon tube was replaced with a new tube, and the needle was removed. Enriched (bound) gametocytes were washed off the column with 4 ml of warm RPMI, the plunger was used to press the last 1 ml of medium through the column. The two 15 ml Falcon tubes, one with the blood mixed to RPMI and the second with the gametocyte suspension, were spun down at 2000 rpm for 5 min in a temperature-controlled centrifuge at 37°C (Eppendorf 5702R). RPMI supernatant was removed from both tubes with 3 ml disposable Pasteur pipettes. To prepare the mosquito blood-meal, the small visible pellet of concentrated gametocytes (~30 µl) was resuspended with 150 µl of prewarmed malaria-naïve serum (Sanquin Bloodbank, Nijmegen, the Netherlands) with a blunt needle and 200 µl of the patients' packed red cells were added and mixed. About 350 µl gametocyte enriched bloodmeal was added to a water-jacket glass feeder as described above.

## Mosquito husbandry and oocyst detection by mercurochrome staining

In both insectaries, at Radboudumc (the Netherlands) and CNRFP (Burkina Faso), following membrane feeding, unfed mosquitoes were immediately removed from cups with an aspirator. On days 4–6 PI, mosquitoes were given a second bloodmeal to synchronize oocyst development. Mosquitoes were kept at 27–29°C in the insectaries on 5–10% glucose and dissected for 7–8 d to assess infection prevalence. Twenty mosquito midguts were stained with 1% mercurochrome and oocysts were examined and confirmed by two independent microscopists at ×400 under an optic microscope (CX 40 Olympus). If oocyst prevalence was above 40%, the infected mosquitoes were transferred to the biosecure insectary in Nijmegen, whilst in Ouagadougou cups with infected mosquitoes were placed into secured metal cages (30 × 30 × 30 cm) and kept in a temperature- and relative humidity-controlled environment (27–29°C and 70–80% HR) with double-screened doors to prevent sporozoite-positive mosquitoes from escaping.

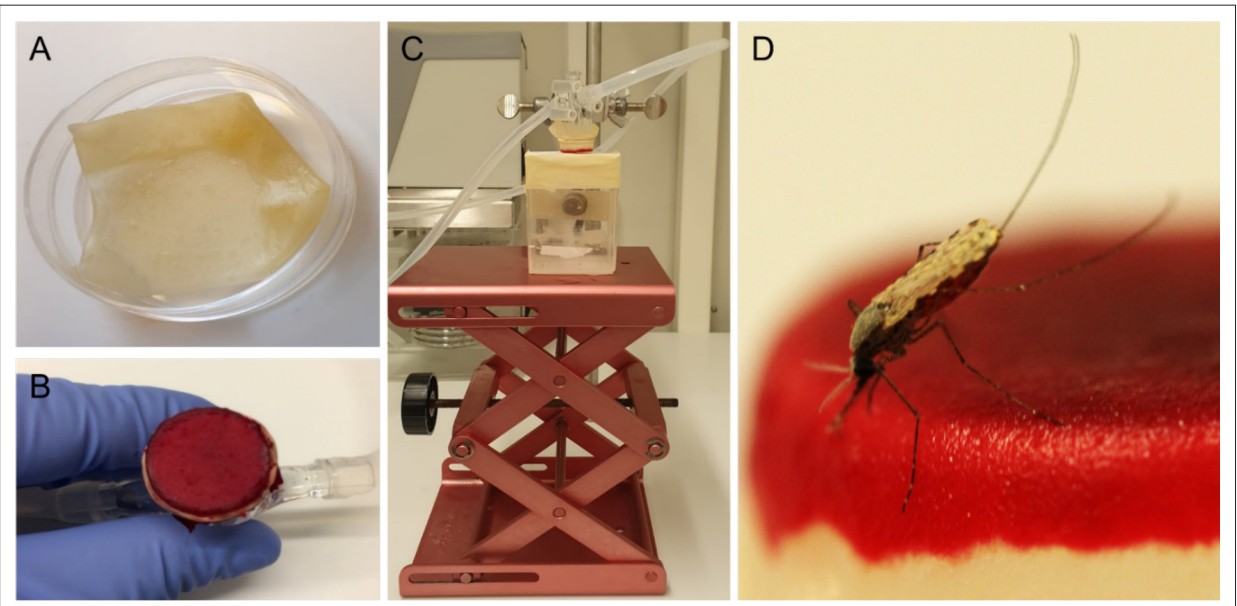

**Figure 6.** Artificial skin feeding procedure. (**A**) Petri dish filled with sterile nuclease-free water, containing the artificial skin. (**B**) Skin folded around a mini-feeder and supplemented with 100 µl of human blood. (**C**) Feeding setup with adjustable table and a small Perspex cage containing a single mosquito underneath. (**D**) A blood-feeding mosquito on artificial skin.

## Extrinsic incubation period

The EIP, defined as the duration of sporogony, was estimated in *An. stephensi* mosquitoes in order to assess (i) the best day to perform the expelling experiment and (ii) if EIP differed in low- vs high-infected mosquitoes. Cages containing 150 mosquitoes were fed with *P. falciparum* NF54 and NF135 gametocytes, and 20 midguts were dissected on days 6–8 PI to assess oocyst infection intensity. Groups of low- (mean of 5 oocysts) and high (mean of 20 oocysts)-infected mosquitoes (prevalence of infection > 70%) were maintained in a secure insectary until salivary gland dissections were performed on days 9–11. Salivary glands for individual mosquitoes were collected in 1.5 ml Eppendorf tubes containing 180 µl oocyst lysis buffer (NaCl 0.1 M: EDTA 25 mM: Tris–HCl 10 mM), and stored at –20°C for further molecular analysis. Mosquito bodies were first homogenized by beat-beating as previously described (*Graumans et al., 2017*) in 100 µl phosphate-buffered saline (PBS).

## Sporozoite-expelling experiments

On 15 and 16 d post feeding, at Radboudumc and CNRFP respectively, infected mosquitoes were used to quantify the number of expelled sporozoites. To prevent contamination, all instruments and equipment were cleaned from nucleic acids by 30 min exposure to sodium hypochlorite (10% in $H_2O$), rinsed with water, and paper-dried on the day before the experiment. New gloves were used each time the experiments were performed. Integra dermal substitute (Dermal Regeneration Template, single layer 20 × 25 cm, ref 68101), hereafter referred to as artificial skin, was cut into 3.5 cm squares (*Figure 6*). Squares were transferred to Petri dishes filled with sterile nuclease-free water (VWR, E476) and left overnight at room temperature (RT). Mosquitoes were individually collected in small Perspex cages (5 × 5 × 7 cm, covered with netting material on the top and bottom sides). Mosquitoes were starved 14–16 hr prior to feeding. On the day of the expelling experiment, artificial skin was transferred with gloves to an inverted positioned glass membrane mini-feeder (convex bottom, 15 mm diameter, ref 70172000) connected to a heated circulating water bath (CORIO C-B5, Julabo) set to 39°C. A rubber band was wrapped around the feeder to secure the artificial skin. Paper tissue was gently pressed on the skin four times to absorb water. Then, 100 µl of naïve donor blood (EDTA, BD Vacutainers, ref 367525) was pipetted on the circular artificial skin area and spread evenly across the surface with the horizontal side of the tip. The feeder was then turned around and placed on top of the cage, without touching the netting, with a maximum of 8 min for mosquito probing. Following mosquito probing, a scalpel (Dalhausen präzisa plus, no. 11) was used to cut the artificial skin above the rubber band around the entire feeder. The artificial skin was transferred with tweezers to a 1.5 ml Eppendorf tube containing 180 µl oocyst lysis buffer, and stored at –20°C. After feeding, mosquitoes were transferred to metal cages. Mosquitoes were kept at 27–29°C in the insectaries on 5% glucose.

## Immunolabeling of intact and ruptured oocysts

Mosquitoes were allowed to digest blood for 3 d to prevent interference with immunolabeling; mosquito salivary glands and midguts were dissected on day 18 PI for *An. stephensi* (Nijmegen), and day 19 PI for *An. coluzzii* (Burkina Faso). Salivary glands were collected in 1.5 ml Eppendorf tubes containing 180 µl oocyst lysis buffer (NaCl 0.1 M: EDTA 25 mM: Tris–HCl 10 mM), and stored at –20°C for further molecular analysis. Midguts were dissected in 20 µl PBS (pH 7.2) without mercurochrome. For experiments performed in Nijmegen, *An. stephensi* midguts were transferred to a fresh drop of (1:400) 3SP2-Alexa 488 anti-CSP antibodies (DyLight 488 Anitibody Labeling Kit, Thermo Fisher Scientific, ref 53024) and incubated for 30 min at RT in a slide humidity incubation box. Following staining, midguts were washed twice with 10 µl of PBS for 10 min. Midguts were transferred to glass slides and secured with a cover slip. Intact/degenerated and ruptured oocysts were counted using an incident light fluorescence microscope GFP filter at 400×. Due to the lack of fluorescence microscopes at CNRFP, we combined a formalin fixation method with immunostaining. *An. coluzzii* midguts were transferred into individual screw cap tubes (Eppendorf) filled with 400 µl of 4% formalin and stored at 4°C until shipped to the Netherlands. Midguts stored in 4% formalin were collected by using a p1000 Gilson pipette with the point of the tip cut and placed on a slide. Midguts were rinsed from formalin three times in PBS 1X-Tween 0.05% by moving the midgut with a needle from drop to drop. They were then transferred to a fresh drop of 3SP2-Alexa 488 anti-CSP antibodies (1:400) and incubated for 20 min. Midguts were rinsed in PBS and examined as described above.

## Sample extraction, sporozoite quantification by qPCR, and amplicon deep sequencing

Serial dilutions of *P. falciparum* sporozoites were generated to prepare standard curves for qPCR. Therefore, pooled salivary glands from highly infected mosquitoes were collected in a glass pestle grinder that contained 500 µl PBS. The sample was homogenized and subsequently diluted 100 times in PBS. Sporozoites were transferred to a hemocytometer and counted under a phase-contrast light microscopy (×400 magnification), by two independent microscopists. Serial dilutions were prepared in PBS using glass test tubes and low binding tips. For each concentration, 100 µl was filled out in a 1.5 ml Eppendorf tube. Eppendorf tubes were stored at –70°C for at least 1 d before sample processing. Prior to DNA extraction, 30 µl proteinase K (QIAGEN, Cat# 19133) was added to Eppendorf tubes containing collected artificial skin and salivary glands samples. PBS was added to all samples to have an equal volume (410 µl) before incubation overnight at 56°C. The following day, total nucleic acids (NA) were extracted with the automated MagNA Pure LC instrument (Roche) using the MagNA Pure LC DNA Isolation Kit – High performance (Roche, product no. 03310515001), and eluted in 50 µl. Samples were used immediately or stored at –20°C. *P. falciparum* sporozoites were quantified by qPCR, targeting the mitochondrial gene COX-1. A previously published primer set (*Boissière et al., 2013*) was modified to improve template annealing, forward primer 5′-CATCAGGAATGTTATTGCTA ACAC-3′ and reverse primer 5′-GGATCTCCTGCAAATGTTGGGTC-3′, resulting in an amplicon length of 112 bp. A probe was designed for amplicon detection 6FAM-ACCGGTTTTAACTGGAGGAGTA-BHQ1. qPCR reactions were prepared with TaqMan Fast Advanced Master Mix (Applied Biosystems, ref 4444557). For each reaction, 12.5 µl mix, 0.4 µl primers (stock 50 µM, final concentration 800 nM), 0.1 µl probe (stock 100 µM, final concentration 400 nM), 7 µl PCR grade water, and 5 µl template DNA were used. In each run, standard curves and negative controls (water) were included. Melt curves were visually inspected. Samples were run on a Bio-Rad CFX 96 real-time System at 95°C for 15 s, followed by 30 cycles of 95°C for 15 s, 60°C for 60 s. To identify unique clones in mosquitoes infected by gametocyte donors with naturally acquired *P. falciparum* gametocytes, samples were genotyped by apical membrane antigen 1 (AMA-1) amplicon sequencing as previously described (*Briggs et al., 2020*).

## Statistical analysis

Statistical analyses were performed in R, version 3.1.12 (*Team RC, 2019*). Associations between $\log_{10}$ oocyst intensity and infection prevalence were modeled using a logistic regression model (using N = 457, *Figure 1B*). The difference in mean $\log_{10}$ oocyst densities between staining types was compared using a *t*-test (*t* = 0.070075, df = 405, p-value=0.9442, N = 406, *Figure 1D*). The association between total sporozoite density and experiment day with the prevalence of salivary gland sporozoite was modeled using a mixed-effects logistic regression with a random intercept for the different experiments (using N = 266, *Figure 2A*). Welch's *t*-test was used to compare sporozoite density between days 9 and 10 (*t* = –2.0467, df = 28.66, p=0.04995, N = 31, *Figure 2B*). The association between total sporozoite load and sporozoite expelling prevalence was modelled using a logistic regression (using N = 186, *Figure 3A*).

Spearman's correlation coefficient $\rho$ was used to assess the association between oocyst sheets and salivary gland sporozoite load (one outlier not included) ( $\rho$ = 0.80, 95% CI, 0.74–0.85; p<0.0001; N = 111, *Figure 3B*); the association between total sporozoite density and the number of sporozoites that was expelled into the artificial skin ( $\rho$ = 0.35, 95% CI, 0.17–0.50; p=0.0002; N = 112, *Figure 3C*) (one outlier not included); the associations between ruptured oocyst density and total sporozoite load, and between total sporozoite load and skin expelling ( $\rho$ = 0.9, 95% CI, 0.80–0.95; p<0.0001; N = 25, *Figure 4C*) and ( $\rho$ = 0.71, 95% CI, 0.52–0.82; p<0.0003; N = 25, *Figure 4D*).

## Acknowledgements

We thank all the study participants for their willingness to support the study and donate blood. We would like to thank all the lab staff, drivers, and procurement for their dedication in the study; Jacqueline Kuhnen, Laura Pelser-Posthumus, Astrid Pouwelsen, and Jolanda Klaassen (Radboudumc) for all mosquito husbandry; Jared Honeycutt and Sophie Maxfield (University of California – San Francisco, USA) for their work on AMA-1 amplicon deep sequencing; Claudia Bin for all the tester units of

IntegraDermal Regeneration Template. Funding was provided by the fellowship from the European Research Council to TB (ERC-CoG 864180; QUANTUM) and an AMMODO Science Award (2019) to TB. PS was supported by the National Institutes of Health through an R01 grant (AI132359).

## Additional information

### Funding

| Funder | Grant reference number | Author |
|---|---|---|
| HORIZON EUROPE European Research Council | ERC-CoG 864180 | Chiara Andolina Wouter Graumans Jordache ramijth Teun Bousema |
| AMMODO Science Award | 2019 | Teun Bousema |
| National Institutes of Health | AI132359 | Photini Sinnis |

The funders had no role in study design, data collection and interpretation, or the decision to submit the work for publication.

### Author contributions

Chiara Andolina, Wouter Graumans, Conceptualization, Formal analysis, Investigation, Methodology, Writing – original draft; Moussa Guelbeogo, Geert-Jan van Gemert, Investigation, Methodology, Writing – review and editing; Jordache Ramijth, Formal analysis, Visualization, Methodology, Writing – review and editing; Soré Harouna, Investigation, Writing – review and editing; Zongo Soumanaba, Writing – review and editing; Rianne Stoter, Marga Vegte-Bolmer, Investigation; Martina Pangos, Resources, Methodology; Photini Sinnis, Conceptualization, Writing – review and editing; Katharine Collins, Sarah G Staedke, Supervision, Writing – review and editing; Alfred B Tiono, Supervision, Investigation, Writing – review and editing; Chris Drakeley, Conceptualization; Kjerstin Lanke, Investigation, Methodology, Writing – original draft; Teun Bousema, Conceptualization, Formal analysis, Supervision, Funding acquisition, Methodology, Writing – original draft, Writing – review and editing

### Author ORCIDs

Wouter Graumans ⓘ https://orcid.org/0000-0003-3952-6491
Geert-Jan van Gemert ⓘ http://orcid.org/0000-0002-2740-5337
Chris Drakeley ⓘ http://orcid.org/0000-0003-4863-075X
Teun Bousema ⓘ http://orcid.org/0000-0003-2666-094X

### Ethics

The study protocol in Burkina Faso was approved by the London School of Hygiene and Tropical Medicine ethics committee (Review number: 14724), the Centre National de Recherche et de Formation sur le Paludisme institutional review board (Deliberation N° 2018/000,002/MS/SG/CNRFP/CIB) and the Ethics Committee for Health Research in Burkina Faso (Deliberation N° 2018-01-010). Donors were recruited at schools in Sapone00E9; Health District, following informed consent. Experiments with in vitro cultured parasites and An. stephensi mosquitoes at Radboud University Medical Center were conducted following approval from the Radboud University Experimental Animal Ethical Committee (RUDEC 2009-019, RUDEC 2009-225).

Reviewer #1 (Public Review): https://doi.org/10.7554/eLife.90989.3.sa1
Reviewer #2 (Public Review): https://doi.org/10.7554/eLife.90989.3.sa2
Reviewer #3 (Public Review): https://doi.org/10.7554/eLife.90989.3.sa3
Author Response https://doi.org/10.7554/eLife.90989.3.sa4

### Data availability

All data generated or analysed during this study are deposited on Dryad: https://doi.org/10.5061/dryad.dbrv15f89.

The following dataset was generated:

| Author(s) | Year | Dataset title | Dataset URL | Database and Identifier |
|---|---|---|---|---|
| Andolina C, Graumans W, Guelbeogo M, Gemert GJV, Ramjith J, Harouna S, Soumanaba Z, Stoter R, Vegte-Bolmer M, Pangos M, Sinnis P, Collins K, Staedke SG, Tiono AB, Drakeley C, Lanke K, Bousema T | 2024 | Data from: Quantification of sporozoite expelling by Anopheles mosquitoes infected with laboratory and naturally circulating *P. falciparum* gametocytes | https://doi.org/10.5061/dryad.dbrv15f89 | Dryad Digital Repository, 10.5061/dryad.dbrv15f89 |

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
