## [Editor Report · eLife assessment]

This **important** study combines experimental infections with laboratory and field *Plasmodium falciparum* isolates to quantify the force of human malaria parasite transmission. By using **compelling** methodological approaches, the authors establish clear positive correlations between mosquito infection levels (as determined by oocyst numbers), sporozoite loads in salivary glands, and sporozoites expelled during feeding. The link between heterogeneous infection levels in the mosquitoes and malaria transmission would be of interest to vector biologists, parasitologists, immunologists, and mathematical modelers.

---

## [Referee Report · Reviewer #1 (Public Review)]

Summary:

There is a long-believed dogma in the malaria field; a mosquito infected with a single oocyst is equally infectious to humans as another mosquito with many oocysts. This belief has been used for goal setting (and modeling) of malaria transmission-blocking interventions. While recent studies using rodent malaria suggest that the dogma may not be true, there was no such study with human *P. falciparum* parasites. In this study, the numbers of oocysts and sporozoite in the mosquitoes and the number of expelled sporozoites into artificial skin from the infected mosquito was quantified individually. There was a significant correlation between sporozoite burden in the mosquitoes and expelled sporozoites. In addition, this study showed that highly infected mosquitoes expelled sporozoites sooner.

Strengths:

• The study was conducted using two different parasite-mosquito combinations; one was lab-adapted parasites with Anopheles stephensi and the other was parasites, which were circulated in infected patients, with An. coluzzii. Both combinations showed statistically significant correlations between sporozoite burden in mosquitoes and the number of expelled sporozoites.

• Usually, this type of study has been done in group bases (e.g., count oocysts and sporozoites at different time points using different mosquitoes from the same group). However, this study determined the numbers in individual bases after multiple optimization and validation of the approach. This individual approach significantly increases the power of correlation analysis.

Weaknesses:

• In a natural setting, most mosquitoes have less than 5 oocysts. Thus, the conclusion is more convincing if the authors perform additional analysis for the key correlations (Fig 3C and 4D) excluding mosquitoes with very high total sporozoite load (e.g., more than 5-oocyst equivalent load).

• As written as the second limitation of the study, this study did not investigate whether all expelled sporozoites were equally infectious. For example, Day 9 expelled sporozoites may be less infectious than Day 11 sporozoites, or expelled sporozoites from high-burden mosquitoes may be less infectious because they experience low nutrient conditions in a mosquito. Ideally, it is nice to test the infectivity by ex vivo assays, such as hepatocyte invasion assay, and gliding assay at least for salivary sporozoites. But are there any preceding studies where the infectivity of sporozoites from different conditions was evaluated? Citing such studies would strengthen the argument.

• Since correlation analyses are the main points of this paper, it is important to show 95%CI of Spearman rank coefficient (not only p-value). By doing so, readers will understand the strengths/weaknesses of the correlations. The p-value only shows whether the observed correlation is significantly different from no correlation or not. In other words, if there are many data points, the p-value could be very small even if the correlation is weak.

---

## [Referee Report · Reviewer #2 (Public Review)]

Summary:

The malaria parasite Plasmodium develops into oocysts and sporozoites inside Anopheles mosquitoes, in a process called sporogony. Sporozoites invade the insect salivary glands in order to be transmitted during a blood meal. An important question regarding malaria transmission is whether all mosquitoes harboring Plasmodium parasites are equally infectious. In this paper, the authors investigated the progression of *P. falciparum* sporozoite development in Anopheles mosquitoes, using a sensitive qPCR method to quantify sporozoites and an artificial skin system to probe for parasite expelling. They assessed the association between oocyst burden, salivary gland infection intensity, and sporozoites expelled.

The data show that higher sporozoite loads are associated with earlier colonization of salivary glands and a higher prevalence of sporozoite-positive salivary glands and that higher salivary gland sporozoite burdens are associated with higher numbers of expelled sporozoites. Intriguingly, there is no clear association between salivary gland burdens and the prevalence of expelling, suggesting that most infections reach a sufficient threshold to allow parasite expelling during a mosquito bite. This important observation suggests that low-density gametocyte carriers, although less likely to infect mosquitoes, could nevertheless contribute to malaria transmission.

Strengths:

The paper is well written and the work is well conducted. The authors used two experimental models, one using cultured *P. falciparum* gametocytes and An. stephensi mosquitoes, and the other one using natural gametocyte infections in a field setup with An. coluzzii mosquitoes. Both studies gave similar results, reinforcing the validity of the observations. Parasite quantification relies on a robust and sensitive qPCR method, and parasite expelling was assessed using an innovative experimental setup based on artificial skin.

Weaknesses:

There is no clear association between the prevalence of sporozoite expelling and the parasite burden. However, high total sporozoite burdens are associated with earlier and more efficient colonization of the salivary glands, and higher salivary gland burdens are associated with higher numbers of expelled sporozoites. While these observations suggest that highly infected mosquitoes could transmit/expel parasites earlier, this is not directly addressed in the study. In addition, whether all expelled sporozoites are equally infectious is unknown. The central question, i.e. whether all infected mosquitoes are equally infectious, therefore remains open.

---

## [Referee Report · Reviewer #3 (Public Review)]

Summary:

This study uses a state-of-the-art artificial skin assay to determine the quantity of *P. falciparum* sporozoites expelled during feeding using mosquito infection (by standardised membrane feeding assay SMFA) using both cultured gametocytes and natural infection. Sporozoite densities in salivary glands and expelled into the skin are quantified using a well-validated molecular assay. These studies show clear positive correlations between mosquito infection levels (as determined by oocyst numbers), sporozoite numbers in salivary glands, and sporozoites expelled during feeding. This indicates potentially significant heterogeneity in infectiousness between mosquitoes with different infection loads and thus challenges the often-made assumption that all infected mosquitoes are equally infectious.

Strengths:

Very rigorously designed studies using very well validated, state-of-the-art methods for studying malaria infections in the mosquito and quantifying load of expelled sporozoites. This resulted in very high-quality data that was well-analyzed and presented. Both sources of gametocytes (cultures vs. natural infection) show consistent results further strengthening the quality of the results obtained.

Weaknesses:

As is generally the case when using SMFAs, the mosquito infections levels are often relatively high compared to wild-caught mosquitoes (e.g. Bombard et al 2020 IJP: median 3-4 ), and the strength of the observed correlations between oocyst sheet and salivary gland sporozoite load even more so between salivary gland sporozoite load and expelled sporozoite number may be dominated by results from mosquitoes with infection levels rarely observed in wild-caught mosquitoes. This could result in an overestimation of the importance of these well-observed positive relationships under natural transmission conditions.

The results obtained from these excellently designed and executed studies very well supported their conclusion - with a slight caveat regarding their application to natural transmission scenarios

This work very convincingly highlights the potential for significant heterogeneity in the infectiousness between individual *P. falciparum*-infected mosquitoes. Such heterogeneity needs to be further investigated and if again confirmed taken into account both when modelling malaria transmission and when evaluating the importance of low-density infections in sustaining malaria transmission.

---

## [Author Response]

We would like to thank the editor and the reviewers for their constructive comments and the chance to revise the manuscript. The suggestions have allowed us to improve our manuscript. We have been able to fulfil all reviewer comments and added new statistical analyses to examine associations for subsets of data. Whilst suggested by a reviewer, we did not perform large-scale experiments to confirm the viability of low sporozoite densities at different time-points post salivary gland colonization. For these assays there are currently no satisfactory in vitro models for sporozoites harvested from single mosquitoes and setting up and validating such experiments could be a PhD project in itself. We do consider this suggestion very relevant but beyond the scope of the current work.

Relevantly, during the time the manuscript was under review at eLife, we have been able to examine the multiplicity of infection in our field experiments. This was, as written in the original manuscript, a key reason to also perform experiments in the field where there is a greater diversity of parasite lines. We have successfully performed AMA-1 amplicon deep sequencing on infected mosquito salivary glands and infected skins. Although this does not change the key messages of the manuscript and is secondary to our main hypothesis, we do consider it a relevant addition since we were able to demonstrate that for some infected mosquitoes from the Burkina Faso study, multiple clones were expelled by mosquitoes during probing on a single piece of artificial skin. We have added a short paragraph to our revised manuscript and updated the acknowledgement section to include the supporting researcher who conducted those experiments.

**Reviewer #1 (Public Review):**
Summary: There is a long-believed dogma in the malaria field; a mosquito infected with a single oocyst is equally infectious to humans as another mosquito with many oocysts. This belief has been used for goal setting (and modelling) of malaria transmission-blocking interventions. While recent studies using rodent malaria suggest that the dogma may not be true, there was no such study with human *P. falciparum* parasites. In this study, the numbers of oocysts and sporozoite in the mosquitoes and the number of expelled sporozoites into artificial skin from the infected mosquito was quantified individually. There was a significant correlation between sporozoite burden in the mosquitoes and expelled sporozoites. In addition, this study showed that highly infected mosquitoes expelled sporozoites sooner.Strengths:The study was conducted using two different parasite-mosquito combinations; one was lab-adapted parasites with Anopheles stephensi and the other was parasites, which were circulated in infected patients, with An. coluzzii. Both combinations showed statistically significant correlations between sporozoite burden in mosquitoes and the number of expelled sporozoites.Usually, this type of study has been done in group bases (e.g., count oocysts and sporozoites at different time points using different mosquitoes from the same group). However, this study determined the numbers in individual bases after multiple optimization and validation of the approach. This individual approach significantly increases the power of correlation analysis.Weaknesses:In a natural setting, most mosquitoes have less than 5 oocysts. Thus, the conclusion is more convincing if the authors perform additional analysis for the key correlations (Fig 3C and 4D) excluding mosquitoes with very high total sporozoite load (e.g., more than 5-oocyst equivalent load).

In the revised manuscript, we have also performed our analysis including only the subset of mosquitoes with low oocyst burden. In our Burkina Faso experiments, where we could not control oocyst density, 48% (15/31) of skins were from mosquitoes with <5 oocyst sheets. Whilst low oocyst densities were thus not very uncommon, we acknowledge that this may have rendered some comparisons underpowered. At the same time, we observe a strong positive trend between oocyst density and sporozoite density and between salivary gland sporozoite density and mosquito inoculum. This makes it very likely that this trend is also present at lower oocyst densities, an association where sporozoite inoculation saturates at high densities is plausible and has been observed before for rodent malaria (DOI: 10.1371/journal.ppat.1008181) whilst we consider it less likely that sporozoite expelling would be more efficient at low (unmeasured) sporozoite densities.

As written as the second limitation of the study, this study did not investigate whether all expelled sporozoites were equally infectious. For example, Day 9 expelled sporozoites may be less infectious than Day 11 sporozoites, or expelled sporozoites from high-burden mosquitoes may be less infectious because they experience low nutrient conditions in a mosquito. Ideally, it is nice to test the infectivity by ex vivo assays, such as hepatocyte invasion assay, and gliding assay at least for salivary sporozoites. But are there any preceding studies where the infectivity of sporozoites from different conditions was evaluated? Citing such studies would strengthen the argument.

We appreciate this thought and can see the value of these experiments. We are not aware of any studies that examined sporozoite viability in relation to the day of salivary gland colonization or sporozoite density.

One previous study assessed the NF54 sporozoite infectivity on different days post infection (days 12-13-14-15-16-18) and observed no clear differences in ‘per sporozoite hepatocyte invasion capacity’ over this period (DOI: 10.1111/cmi.12745). We nevertheless agree that it is conceivable that sporozoites require maturation in the salivary glands and might not all be equally infectious. While hepatocyte invasion experiments are conducted with bulk harvesting of all the sporozoites that are present in the salivary glands, it would even be more interesting to assess the invasion capacity of the smaller population of sporozoites that migrate to the proboscis to be expelled. This would, as the reviewer will appreciate, be a major endeavour. To do this well the expelled sporozoites would need to be harvested from the salivary glands/proboscis and used in the best and most natural environment for invasion. The suggested work would thus depend on the availability of primary hepatocytes since conventional cell-lines like HC-04 are likely to underestimate sporozoite invasion. Importantly, there are currently no opportunities to include the barrier of the skin environment in invasion assays whilst this may be highly important in determining the likelihood that sporozoites manage to achieve invasion and give rise to secondary infections. In short, we agree with the reviewer that these experiments are of interest but consider these well beyond the scope of the current work. We have added a section to the Discussion section to highlight these future avenues for research. ‘Of note, our assessments of EIP and of sporozoite expelling did not confirm the viability of sporozoites. Whilst the infectivity of sporozoites at different time-points post infection has been examine previously (https://doi.org/10.1111/cmi.12745), these experiments have never been conducted with individual mosquito salivary glands. To add to this complexity, such experiments would ideally retain the skin barrier that may be a relevant determinant for invasion capacity and primary hepatocytes.’

Since correlation analyses are the main points of this paper, it is important to show 95% CI of Spearman rank coefficient (not only p-value). By doing so, readers will understand the strengths/weaknesses of the correlations. The p-value only shows whether the observed correlation is significantly different from no correlation or not. In other words, if there are many data points, the p-value could be very small even if the correlation is weak.

We appreciate this comment and agree that this is indeed insightful. We have added the 95% confidence intervals to all figure legends and main text. We also provide them below.

Fig 3b: 95% CI: 0.74, 0.85

Fig 3c: 95% CI: 0.17, 0.50

Fig 4c: 95% CI: 0.80, 0.95

Fig 4d: 95% CI: 0.52, 0.82

Supp Fig 5a: 95% CI: 0.74, 0.85

Supp Fig 5b: 95% CI: 0.73, 0.93

Supp Fig 6: 95% CI: 0.11, 0.48

Supp Fig 7: 95% CI: -0.12, 0.16

**Reviewer #2 (Public Review):**
Summary: The malaria parasite Plasmodium develops into oocysts and sporozoites inside Anopheles mosquitoes, in a process called sporogony. Sporozoites invade the insect salivary glands in order to be transmitted during a blood meal. An important question regarding malaria transmission is whether all mosquitoes harbouring Plasmodium parasites are equally infectious. In this paper, the authors investigated the progression of *P. falciparum* sporozoite development in Anopheles mosquitoes, using a sensitive qPCR method to quantify sporozoites and an artificial skin system to probe for parasite expelling. They assessed the association between oocyst burden, salivary gland infection intensity, and sporozoites expelled.The data show that higher sporozoite loads are associated with earlier colonization of salivary glands and a higher prevalence of sporozoite-positive salivary glands and that higher salivary gland sporozoite burdens are associated with higher numbers of expelled sporozoites. Intriguingly, there is no clear association between salivary gland burdens and the prevalence of expelling, suggesting that most infections reach a sufficient threshold to allow parasite expelling during a mosquito bite. This important observation suggests that low-density gametocyte carriers, although less likely to infect mosquitoes, could nevertheless contribute to malaria transmission.Strengths: The paper is well written and the work is well conducted. The authors used two experimental models, one using cultured *P. falciparum* gametocytes and An. stephensi mosquitoes, and the other one using natural gametocyte infections in a field setup with An. coluzzii mosquitoes. Both studies gave similar results, reinforcing the validity of the observations. Parasite quantification relies on a robust and sensitive qPCR method, and parasite expelling was assessed using an innovative experimental setup based on artificial skin.Weaknesses: There is no clear association between the prevalence of sporozoite expelling and the parasite burden. However, high total sporozoite burdens are associated with earlier and more efficient colonization of the salivary glands, and higher salivary gland burdens are associated with higher numbers of expelled sporozoites. While these observations suggest that highly infected mosquitoes could transmit/expel parasites earlier, this is not directly addressed in the study. In addition, whether all expelled sporozoites are equally infectious is unknown. The central question, i.e. whether all infected mosquitoes are equally infectious, therefore remains open.

We agree that the manuscript provides important steps forward in our understanding of what makes an infectious mosquito but does not conclusively demonstrate that highly infected mosquitoes are more likely to initiate a secondary infection. We consider this to be beyond the scope of the current work although the current work lays the foundation for these important future studies. For human Plasmodium infections the most satisfactory answer on the infectiousness of low versus high infected mosquitoes comes from controlled human infection models. In response to reviewer comments, we have extended our Discussion section to highlight this importance. To accommodate the (very fair) reviewer comments, we have avoided any phrasings that suggest that our findings demonstrate differences in transmission.

**Reviewer #3 (Public Review):**
Summary: This study uses a state-of-the-art artificial skin assay to determine the quantity of *P. falciparum* sporozoites expelled during feeding using mosquito infection (by standardised membrane feeding assay SMFA) using both cultured gametocytes and natural infection. Sporozoite densities in salivary glands and expelled into the skin are quantified using a well-validated molecular assay. These studies show clear positive correlations between mosquito infection levels (as determined by oocyst numbers), sporozoite numbers in salivary glands, and sporozoites expelled during feeding. This indicates potentially significant heterogeneity in infectiousness between mosquitoes with different infection loads and thus challenges the often-made assumption that all infected mosquitoes are equally infectious.Strengths: Very rigorously designed studies using very well validated, state-of-the-art methods for studying malaria infections in the mosquito and quantifying load of expelled sporozoites. This resulted in very high-quality data that was well-analyzed and presented. Both sources of gametocytes (cultures vs. natural infection) show consistent results further strengthening the quality of the results obtained.Weaknesses: As is generally the case when using SMFAs, the mosquito infections levels are often relatively high compared to wild-caught mosquitoes (e.g. Bombard et al 2020 IJP: median 3-4 ), and the strength of the observed correlations between oocyst sheet and salivary gland sporozoite load even more so between salivary gland sporozoite load and expelled sporozoite number may be dominated by results from mosquitoes with infection levels rarely observed in wild-caught mosquitoes. This could result in an overestimation of the importance of these well-observed positive relationships under natural transmission conditions. The results obtained from these excellently designed and executed studies very well supported their conclusion - with a slight caveat regarding their application to natural transmission scenarios

For efficiency and financial reasons, we have worked with an approach to enhance mosquito infection rates. If we had worked with gametocytes at physiological concentrations and a small number of donors, we probably have had considerably lower mosquito infection rates. Whilst this would indeed result in lower infection burdens in the sparse infected mosquitoes, addressing the reviewer concern, it would have made the experiments highly inefficient and expensive. The skin mimic was initially provided free of charge when the matrix was close to the expiry date but for the experiments in Burkina Faso we had to purchase the product at market value. Whilst we consider the biological question sufficiently important to justify this investment – and think our findings prove us right – it remained important to avoid using skins for uninfected mosquitoes. Since oocyst prevalence and density are strongly correlated (doi: 10.1016/j.ijpara.2012.09.002; doi: 10.7554/eLife.34463), a low oocyst density in natural infections typically coincides with a high proportion of negative mosquitoes.

Of note, our approach did result in the inclusion of 15 skins from infected mosquitoes with 1-4 oocysts. This number may be modest but we did include observations from this low oocyst range which is, we agree, highly important for better understanding malaria epidemiology.

This work very convincingly highlights the potential for significant heterogeneity in the infectiousness between individual *P. falciparum*-infected mosquitoes. Such heterogeneity needs to be further investigated and if again confirmed taken into account both when modelling malaria transmission and when evaluating the importance of low-density infections in sustaining malaria transmission.

**Reviewer #4 (Public Review):**
Summary: The study compares the number of sporozoites expelled by mosquitoes with different Plasmodium infection burden. To my knowledge this is the first report comparing the number of expelled *P. falciparum* sporozoites and their relation to oocyst burden (intact and ruptured) and residual sporozoites in salivary glands. The study provides important evidence on malaria transmission biology although conclusions cannot be drawn on direct impact on transmission.Strengths: Although there is some evidence from malaria challenge studies that the burden of sporozoites injected into a host is directly correlated with the likelihood of infection, this has been done using experimental infection models which administer sporozoites intravenously. It is unclear whether the same correlation occurs with natural infections and what the actual threshold for infection may be. Host immunity and other host related factors also play a critical role in transmission and need to be taken into consideration; these have not been mentioned by the authors. This is of particular importance as host immunity is decreasing with reduction in transmission intensity.Weaknesses: The natural infections reported in the study were not natural as the authors described. Gametocyte enrichment was done to attain high oocyst infection numbers. Studying natural infections would have been better without the enrichment step. The infected mosquitoes have much larger infection burden than what occurs in the wild.Nevertheless, the findings support the same results as in the experiments conducted in the Netherlands and therefore are of interest. I suggest the authors change the wording. Rather than calling these "natural" infections, they could be called, for example, "experimental infections with wild parasite strains".

We have addressed these concerns and, in the process, also changed our manuscript title. The following sentences have been changed:

“It is currently unknown whether all *Plasmodium falciparum* infected mosquitoes are equally infectious. We assessed sporogonic development using cultured gametocytes in the Netherlands and natural infections in Burkina Faso”.

Now reads:“It is currently unknown whether all *Plasmodium falciparum* infected mosquitoes are equally infectious. We assessed sporogonic development using cultured gametocytes in the Netherlands and experimental infections with naturally circulating parasite strains in Burkina Faso”.226-228 “Experimental infections with naturally circulating parasite strains show comparable correlation between oocyst density, salivary gland density and sporozoite inoculum”.

Has now replaced the original phrasing:“Natural infected mosquitoes by gametocyte carriers in Burkina Faso show comparable correlation between oocyst density, salivary gland density and sporozoite inoculum”.

I do not believe the study results generate sufficient evidence to conclude that lower infection burden in mosquitoes is likely to result in changes to transmission potential in the field. In study limitations section, the authors say "In addition, our quantification of sporozoite inoculum size is informative for comparisons between groups of high and low-infected mosquitoes but does not provide conclusive evidence on the likelihood of achieving secondary infections. Given striking differences in sporozoite burden between different Plasmodium species - low sporozoite densities appear considerably more common in mosquitoes infected with P. yoelii and P. berghei the association between sporozoite inoculum and the likelihood of achieving secondary infections may be best examined in controlled human infection studies. However, in the abstract conclusion the authors state "Whilst sporozoite expelling was regularly observed from mosquitoes with low infection burdens, our findings indicate that mosquito infection burden is associated with the number of expelled sporozoites and may need to be considered in estimations of transmission potential." Kindly consider ending the sentence at "expelled sporozoites." Future studies on CHMI can be recommended as a conclusion if authors feel fit.

We agree that we need to be very cautious with conclusions on the impact of our findings for the infectious reservoir. We have rephrased parts of our abstract and have updated the Discussion section following the reviewer suggestions. We agree with the reviewer that CHMI studies are recommended and have expanded the Discussion section to make this clearer. The sentence in the abstract now ends as:

"Whilst sporozoite expelling was regularly observed from mosquitoes with low infection burdens, our findings indicate that mosquito infection burden is associated with the number of expelled sporozoites. Future work is required to determine the direct implications of these findings for transmission potential."

**Reviewer #1 (Recommendations For The Authors):**
• Prevalence data shown in Fig 2A and Table S1 are different. For example, >50K at Day 11, Fig 2A shows ~85% prevalence, but Table S1 says 100%. If the prevalence in Table S1 shows a proportion of observations with positive expelled sporozoites (instead of a proportion of positive mosquitoes shown in Fig 2A), then the prevalence for <1K at Day 11 cannot be 6.7% (either 0 or 20% as there were a total of 5 observations). So in either case, it is not clear why the numbers shown in Fig 2A and Table S1 are different.

Figure 2A and Table S2 are estimated prevalence and odds ratios from an additive logistic regression model (i.e. excluding the interaction between day and sporozoite categories). Table S1 includes this interaction when estimating prevalence and odds ratios and as we can see some categories in the interaction were extremely small resulting in blown up confidence intervals especially in day 11. So Table S1 and Fig 2A are the results from two different models. Whilst our results are thus correct, we can understand the confusion and have added a sentence to explain the model used in the figure/table legends.

Figure. 2 Extrinsic Incubation Period in high versus low infected mosquitoes. A. Total sporozoites (SPZ) per mosquito in body plus salivary glands (x-axis) were binned by infection load <1k; 1k-10k; 10k-50k; >50k and plotted against the proportion of mosquitoes (%) that were sporozoite positive (y-axis) as estimated from an additive logistic regression model with factors day and SPZ categories.Supplementary Table S1. The extrinsic incubation period of *P. falciparum* in An. stephensi estimated by quantification of sporozoites on day 9, 10, 11 by qPCR. Based on infection intensity mosquitoes were binned into four categories (<1k, 1k-10k, 10k-50k, >50) that was assessed by combining sporozoite densities in the mosquito body and salivary gland. Prevalences and odds ratios were estimated from a logistic regression model with factors day, SPZ category and their interaction.

There are 3 typos in the paper. Please fix them.Line 464; ...were counted using a using an incident....Line 473; Supplementary Figure 7 should be Fig S8.Line 508: ...between days 9 and 10 using a (t=-2.0467)....

We appreciate the rigour in reviewing our text and have corrected all typos.

**Reviewer #2 (Recommendations For The Authors):**
High infection burdens may result in earlier expelling capacity in mosquitoes, which would reflect more accurately the EIP. The fact that earlier colonization of SG and correlation between SG burden and numbers expelled suggest it could be the case, but it would be interesting to directly measure the prevalence of expelling over time to directly assess the effect of the sporozoite burden (not just at day 15 but before). This could reveal how the parasite burden in mosquitoes is a determinant of transmission.

We appreciate this suggestion and will consider this for future experiments. It adds another variable that is highly relevant but will also complicate comparisons where sporozoite expelling is related to both time since infectious blood meal and salivary gland sporozoite density (that is also dependent on time since infectious bloodmeal). Moreover, we then consider it important to measure this over the entire duration of sporozoite expelling, including late time-points post infectious bloodmeal. This may form part of a follow-up study.

Another question is whether all sporozoites (among expelled parasites) are equally infective, i.e. susceptible to induce secondary infection. If not, this could reconcile the data of this study and previous results in the rodent model where high burdens were associated with an increased probability to transmit.

As also indicated above, we are aware of a single study that assessed NF54 sporozoite infectivity on different days post infection (days 12-13-14-15-16-18) and observed no clear differences in ‘per sporozoite hepatocyte invasion capacity’ over this period (DOI: 10.1111/cmi.12745). We nevertheless agree that it is conceivable that sporozoites require maturation in the salivary glands and might not all be equally infectious. While hepatocyte invasion experiments are conducted with bulk harvesting of all the sporozoites that are present in the salivary glands, it would even be more interesting to assess the invasion capacity of the smaller population of sporozoites that migrate to the proboscis to be expelled. This would, as the reviewer will appreciate, be a major endeavour. To do this well the expelled sporozoites would need to be harvested from the salivary glands/proboscis and used in the best and most natural environment for invasion. The suggested work would thus depend on the availability of primary hepatocytes since conventional cell-lines like HC-04 are likely to underestimate sporozoite invasion. Importantly, there are currently no opportunities to include the barrier of the skin environment in invasion assays whilst this may be highly important in determining the likelihood that sporozoites manage to achieve invasion and give rise to secondary infections. In short, we agree with the reviewer that these experiments are of interest but consider these well beyond the scope of the current work. We have added a section to the Discussion section to highlight these future avenues for research. ‘Of note, our assessments of EIP and of sporozoite expelling did not confirm the viability of sporozoites. Whilst the infectivity of sporozoites at different time-points post infection has been examine previously (ref), these experiments have never been conducted with individual mosquito salivary glands. To add to this complexity, such experiments would ideally retain the skin barrier that may be a relevant determinant for invasion capacity and primary hepatocytes.’

The authors evaluated oocyst rupture at day 18, i.e. 3 days after feeding experiments (performed at day 15). Did they check in control experiments that the prevalence of rupture oocysts does not vary between day 15 and day 18?

We did not do this and consider it very unlikely that there is a noticeable increase in the number of ruptured oocysts between days 15 and 18. We observe that salivary gland invasion plateaus around day 12 and the provision of a second bloodmeal that is known to accelerate oocyst maturation and rupture (doi: 10.1371/journal.ppat.1009131) makes it even less likely that a relevant fraction of oocysts ruptures very late. Perhaps most compellingly, the time of oocyst rupture will depend on nutrient availability and rupture could thus occur later for oocysts from a heavily infected gut compared to oocysts from mosquitoes with a low infection burden. We observe a very strong association between salivary gland sporozoite density (day 15) and oocyst density (assessed at day 18) without any evidence for change in the number of sporozoites per oocyst for different oocyst densities. In our revised manuscript we have also assessed correlations for different ranges of oocyst intensities and see highly consistent correlation coefficients and find no evidence for a change in ‘slope’. If oocyst rupture would regularly happen between days 15 and 18 and this late rupture would be more common in heavily infected mosquitoes, we would expect this to affect the associations presented in figures 3B and 4C This is not the case.

The authors report higher sporozoite numbers per oocyst and a higher proportion of SG invasion as compared to previous studies (30-50% rather than 20%). How do they explain these differences? Is it due to the detection method and/or second blood meal? Or parasite species?

We were also intrigued by these findings in light of existing literature. To address potential discrepancies, it is indeed possible that the 2nd bloodmeal made a difference. In addition, NF54 is known to be a highly efficient parasite in terms of gametocyte formation and transmission. And there are marked differences in these performances between NF54 isolates and definitely between NF54 and its clone 3D7 that is regularly used. We also used a molecular assay to detect and quantify sporozoites but consider it less likely that this is a major factor in terms of explaining SG invasion since sporozoite densities were typically within the range that would be detected by microscopy. We can only hypothesize that the 2nd bloodmeal may have contributed to these findings and acknowledge this in the revised Discussion section.

The median numbers of expelled sporozoites seem to be higher in the natural gametocyte infection experiments as compared to the cultures. Is it due to the mosquito species (An. coluzzii versus An. stephensi?).

The added value of our field experiments, a more relevant mosquito species and more relevant parasite isolates, is also a weakness in terms of understanding possible differences between in vitro experiments and field experiments with naturally circulating parasite strains. We only conclude that our in vitro experiments do not over-estimate sporozoite expelling by using a highly receptive mosquito source and artificially high gametocyte densities. We have clarified this in the revised Discussion.

39% of sporozoite-positive mosquitoes failed to expel, irrespective of infection densities. Could the authors discuss possible explanations for this observation?

In paragraph 304-307 we now write that:” This finding broadly aligns with an earlier study of Medica and Sinnis that reported that 22% of P. yoelii infected mosquitoes failed to expel sporozoites. For highly infected mosquitoes, this inefficient expelling has been related to a decrease of apyrase in the mosquito saliva”.

In Figure 3, it would be interesting to zoom in the 0-1k window, below the apparent threshold for successful expelling.

We have generated correlation estimates for different ranges of oocyst and sporozoite densities and added these in Supplementary Table 5. We agree that this helps the reader to appreciate the contribution of different ranges of parasite burden to the observed associations.

In Fig S8. Did they observe intact oocysts with fixed samples? These could be shown as well in the figure.

We have incorporated this comment. An intact oocyst from fixed samples was now added to Fig S10.

Minor points-line 119: LOD and LOQ could be defined here.

We agree that this should have been defined. We changed line 119 to explain LOD and LOQ to: …“the limit of detection (LOD) and limit of quantification (LOQ)”….

line 126: the title does not reflect the content of this paragraph.

We have changed the title: “Immunolabeling allows quantification of ruptured oocysts ”into: A comparative analysis of oocyst densities using mercurochrome staining and anti-CSP immunostaining.

-line 269: infectivity is not appropriate. The data show colonization of SG.

Line 269: infectivity has been changed with colonization of salivary glands.

There seems to be a problem with Fig S6. The graph seems to be the same as Fig 3C. Please check whether the graph and legends are correct.

Supplementary Figure 6 shows the sporozoite expelling density in relation to infection burden with a threshold set at > 20 sporozoites while Fig 3C shows the total sporozoite density (residual salivary gland sporozoites + sporozoites expelled, X-axis) in relation to the number of expelled sporozoites (Y-axis) by COX-1qPCR without any threshold density. We have explained this in more detail in the revised supplemental figure where we now state

“Of note, this figure differs from Figure 3C in the main text in the following manner. This figure presents sporozoite expelling density in relation to infection burden with a threshold set at > 20 sporozoites to conclude sporozoite positivity while Figure 3C shows the total sporozoite density (residual salivary gland sporozoites + sporozoites expelled, X-axis) in relation to the number of expelled sporozoites (Y-axis) by COX-1 qPCR without any threshold density and thus includes all observations with a qPCR signal”

**Reviewer #3 (Recommendations For The Authors):**
Congratulations to the authors for the really excellently designed and rigorously conducted studies.My main concern is in regards to the relatively high oocyst numbers in their experimental mosquitoes (from both sources of gametocytes) compared to what has been reported from wild-caught mosquitoes in previous studies in Burkina Faso.

We have addressed this concern above. For completeness, we include the main points here again. We enriched gametocytes for efficiency reasons, experiments on gametocytes at physiological concentrations would have resulted in a lower oocyst density and thus more ‘natural’ although a minority of individuals achieves very high oocyst densities in all studies that included a broad range of oocyst densities e.g. doi: 10.1016/j.exppara.2014.12.010; doi: 10.1016/S1473-3099(18)30044-6. Of note, we did include 15 skins from low oocyst densities (1-4 oocysts). Whilst low oocyst densities were thus not very uncommon in our sample set, we acknowledge that this may have rendered some comparisons underpowered. At the same time, we observe a strong positive trend between oocyst density and sporozoite density and between salivary gland sporozoite density and mosquito inoculum. This makes it very likely that this trend is also present at lower oocyst densities, an association where sporozoite inoculation saturates at high densities is plausible and has been observed before for rodent malaria (DOI: 10.1371/journal.ppat.1008181) whilst we consider it less likely that sporozoite expelling would be more efficient at low (unmeasured) sporozoite densities. In the revised manuscript we have also performed our analysis including only the subset of mosquitoes with low oocyst burden.

The best way to address this would be to do comparable artificial skin-feeding experiments on such wild-caught mosquitoes, but I appreciate that this is very difficult to do.

This would indeed by difficult to do. Mostly because infection status can only be examined post-hoc and it is likely that >95% of mosquitoes are sporozoite negative at the moment experiments are conducted (in many settings this will even be >99%). Importantly, also in wild-caught mosquitoes very high oocyst burdens are observed in a small but relevant subset of mosquitoes (doi: 10.1016/j.ijpara.2020.05.012).

Instead, I would suggest the authors conduct addition analysis of their data using different cut-offs for maximum oocyst numbers (e.g. <5, <10, <20) to determine if these correlations hold across the entire range of observed oocyst sheets and salivary gland sporozoite load.

We have provided these calculations for the proposed range of oocyst numbers. In addition, we also provided them for a range of sporozoite densities. These findings are now provided in

Entire range of observed oocyst sheets and salivary gland sporozoite load. A minor point on the regression lines in Figures 3 & 4: both variables in these plots have inherent variation (measurement & natural), but regression techniques such as reduced major exit regression (MAR) that allow error in both x and y variables may be preferable to a standard lines regression. Also, as it is implausible that mosquitoes with zero sporozoite in salivary glands expel several hundred sporozoites at feeding, the regression should probably also be constrained to pass through the 0,0 point.

Since the main priority of the analyses is the correlation, and not the fit of the regression line – which is only for indication, and also because of the availability of software, we did not change the type of regression. We have however added a disclaimer to the legend, and we have also forced the intercept to 0 – which does indeed better reflect the biological association. Additionally we added 95% confidence intervals to all Spearman’s correlation coefficients in the legends.